# Spatial exosome analysis using cellulose nanofiber sheets reveals the location heterogeneity of extracellular vesicles

Akira Yokoi [1,2,3,13] ✉, Kosuke Yoshida [1,2], Hirotaka Koga [3,4], Masami Kitagawa[1,5], Yukari Nagao[1], Mikiko Iida[6], Shota Kawaguchi[6], Min Zhang[6], Jun Nakayama [7,8], Yusuke Yamamoto[7], Yoshinobu Baba[6,9,10], Hiroaki Kajiyama [1] & Takao Yasui [2,6,9,11,12,13] ✉

Extracellular vesicles (EVs), including exosomes, are recognized as promising functional targets involved in disease mechanisms. However, the intravital heterogeneity of EVs remains unclear, and the general limitation for analyzing EVs is the need for a certain volume of biofluids. Here, we present cellulose nanofiber (CNF) sheets to resolve these issues. We show that CNF sheets capture and preserve EVs from ~10 μL of biofluid and enable the analysis of bioactive molecules inside EVs. By attaching CNF sheets to moistened organs, we collect EVs in trace amounts of ascites, which is sufficient to perform small RNA sequence analyses. In an ovarian cancer mouse model, we demonstrate that CNF sheets enable the detection of cancer-associated miRNAs from the very early phase when mice did not have apparent ascites, and that EVs from different locations have unique miRNA profiles. By performing CNF sheet analyses in patients, we identify further location-based differences in EV miRNA profiles, with profiles reflecting disease conditions. We conduct spatial exosome analyses using CNF sheets to reveal that ascites EVs from cancer patients exhibit location-dependent heterogeneity. This technique could provide insights into EV biology and suggests a clinical strategy contributing to cancer diagnosis, staging evaluation, and therapy planning.

Extracellular vesicles (EVs), including exosomes, are actively secreted from all living cells and play an essential role in cell-to-cell communication[1–3]. The biological characteristics of EVs may reflect their cellular origin, function and biogenesis, and therefore, EVs in body fluids my reflect a large diversity of the total EV production from cells in a wide variety of tissues[4,5]. In regard to human bodies, there are many types of fluids carrying EVs, and EV heterogeneity can be further complicated[6,7]. In cancer biology, cellular heterogeneity in the tumor microenvironment has been well recognized, and numerous studies have been designed to address this complexity in recent decades[8]. Elucidating subpopulations of EVs is also now an essential task[9], but actual EV heterogeneity in the human body has been impossible to

investigate because EVs are constantly circulating in body fluids. In general, many cancer patients with solid tumors, such as gastrointestinal or gynecological cancers, have tumors in the abdominal cavity, and in some cases, cancerous ascites accumulates in these patients[10]. EVs in ascites have multiple functions in tumor progression or immune maintenance, and they should be a key target to develop therapeutic strategies[11,12]. From a theoretical point of view, ascites-EV near primary tumors, on the surface of the liver, or in the pelvic cavity might have different molecular profiles, and this location-based heterogeneity has not been studied. To elucidate this aspect, it is necessary to innovate a new modality because we did not have ways to obtain such EVs in actual human bodies.

Recently it is known that numerous ways to obtain EVs from body fluids, such as serial centrifugation, density gradient centrifugation, or size-exclusion chromatography exist, and moreover, target specific technology to measure EVs, such as lipid patch microarrays or immune-capturing methods are also applicable if the target molecule to be analyzed has been determined[13]. One of the major challenges of EV isolation is the need for certain sample volumes for analysis, and the conventional ultracentrifugation method cannot be used to analyze microvolume body fluids. To remove these technical barriers, we propose the use of cellulose nanofiber paper (called nanopaper) to directly capture EVs from trace amounts of patient body fluids even on moist organs during surgery and stably store the EVs until postsurgery analyses. The reason why we chose the material of cellulose because it shares the same polysaccharide backbone as sterilized gauze, which has a demonstrated history of successful application inside the body. Cellulose nanofibers, which mainly originate from wood cell walls, have attracted considerable attention owing to their fascinating properties, such as low density (1.6 g/cm$^3$), high strength (2–3 GPa), large specific surface areas (up to ca. 800 m$^2$/g), abundant hydroxyl groups, strong water absorption, sustainability, biocompatibility, and biodegradability[14–16]. Additionally, cellulose nanofibers can now be produced on a large-scale in industrial operations, and their market is projected to reach USD 2.0 billion by 2030[17]. Starting from a cellulose nanofiber/water dispersion, cellulose nanofibers can be used to fabricate nanopaper with tailorable nanostructures by controlling the packing behaviors of cellulose nanofibers during drying[18–21]. The packing behaviors of cellulose nanofibers and the resulting pore size distribution of the nanopaper can be tuned by the surface tension of a surrounding liquid before drying[20], thereby affording intentional opening/closing of the porous nanostructures of the nanopaper. When the nanopaper was prepared by drying in the presence of water with a high surface tension, the resulting nanopaper had densely packed nanostructures (closed pores), providing oxygen barrier properties[22,23]. Nanopaper with porous nanostructures (opened pores) was also prepared by drying in the presence of a low-surface-tension liquid, such as *tert*-butyl alcohol (*t*-BuOH)[18,20,21], and has been applied to aerosol treatment[21] and water/wastewater treatment processes[24]. To the best of our knowledge, however, no one has developed capturing EVs with sizes ca. 40–200 nm within porous nanopaper. We conceived of using the biocompatibility and open pores of the porous nanopaper to directly capture EVs from body fluids on human tissue, and using the closed pores after drying and the resulting oxygen barrier properties to store the captured EVs.

In this study, the porous nanopaper is tailored, denoted CNF sheet, and used for capturing EVs from body fluids on human tissues directly and for storing the EVs. We design the pore sizes based on the small EV size to be capture and the paper thickness based on the balance between handleability and adhesion to human tissues. CNF sheet can capture EVs from 10 μL of body fluids by simply attaching to the moistened tissues via capillary force. When the CNF sheet is dried, the porous nanostructures close automatically due to the high surface tension of water. After drying out, the CNF sheet with closed pores can also store the captured EVs at room temperature for at least 7 days, and the captured EVs within the sheet can be released in phosphate buffer. By using CNF sheets, various EVs are captured and those EVs are mainly CD63-positive small EVs. In addition, small RNAs in EVs can be identified by using CNF sheet capture for small RNA profile analysis. Through CNF sheet analysis in the body, this method unveils the unique profile of ascites EVs in the very early phases of cancer progression in vivo and the actual location-based heterogeneity of EVs in patients. This spatial exosome analysis can provide new concepts of EV isolation/preservation and EV translation biology in cancer.

## Results

### CNF sheets capture intact EVs

The EV capture and storage process using the CNF sheet consists of the following steps: EV capture by absorption of body fluid, EV storage by drying, washing, and EV release (Fig. 1a, Supplementary movie 1). When 10 μL of body fluid is dropped onto the CNF sheet, the EVs in the fluid penetrate to the inside of the sheet owing to its tailored pores and sufficient water absorbency (Supplementary Fig. 1). Subsequent drying treatment closes the pores within the CNF sheet by the aggregation of cellulose nanofibers, allowing the EVs to be stored at room temperature for 7 days; after 7 days, a 10-s washing with PBS opens the pores slightly, but not enough to release the EVs inside, and removes any contaminants adsorbed on the surface {field-emission scanning electron microscope (FE-SEM) images in Fig. 1a}. Finally, EVs can be recovered from the fully opened pores within the CNF sheet by immersing it in PBS for 5 min. Note that the FESEM images shown in Fig. 1a represent the results obtained through saliva supply. The CNF sheet fabricated in this study was ~3 inches in diameter and translucent at the time of fabrication due to the pore sizes of ~300 nm but became more transparent after supplying body fluid and then drying out due to the closed pores (Fig. 1b). The pore size distributions of the CNF sheet also varied with each operation: ~300 nm at fabrication, ~10 nm after drying out (Supplementary Fig. 2), ~20 nm after washing, and ~300 nm after EV recovery (Fig. 1c). Such a series of changes in pore size distributions within the CNF sheet can be associated with drying/wetting-induced pore closing/opening because of the interactions of water and the abundant hydroxyl groups of cellulose nanofibers, as follows: (1) The open pores within the CNF sheet close after the addition of body fluid and then drying because the evaporation of water, which has a high surface tension, causes aggregation of cellulose nanofibers and the formation of hydrogen bonds between the nanofibers. During this stage, the EVs form hydrogen bonds with the hydroxyl groups of the nanofibers (Supplementary Fig. 3). (2) The closed pores gradually open after immersion in water because water can penetrate into the hydrogen bonds between cellulose nanofibers and thus gradually widen the distance between the nanofibers. During this stage, the hydrogen bonds between the EVs and the nanofibers are broken due to the hydration of the EVs, resulting in the release of the EVs from the nanofibers (Supplementary Fig. 3). The CNF sheet had sufficient wet strength to be used in each operation.

Before using the CNF sheet to directly capture EVs from moistened organs, the concept of EV capture in serum was tested. After 10 μL of serum was applied to the CNF sheet, the CNF sheet was dried, stored, and washed, and the recovered EVs had a size distribution of 40-300 nm (Fig. 1d). Comparing the size distribution of contaminants from 0.22 μm-filtered PBS samples (Fig. 1e, Supplementary Fig. 4a), we can say that the CNF sheet was able to isolate EVs from serum samples. In a comparison of the concentration of EVs recovered from 10 μL serum with contaminants from 0.22 μm-filtered PBS samples, the CNF sheets recovered an EV concentration of ~10$^{11}$ particles/mL (Fig. 1e). The amount of EVs introduced versus the amount recovered showed an ~100% recovery efficiency up to 10$^9$ particles (Supplementary Fig. 5). Cryo-electron microscopy (cryo-EM) showed that the EVs recovered from 10 μL serum had spherical shapes covered with lipid bilayers (Fig. 1f), while retaining their original EV size. (Supplementary Fig. 6), and we also confirmed membrane proteins as EV markers (Fig. 1g, h). Based on this characterization, the EVs recovered from 10 μL serum using the CNF sheet were categorized as CD63-positive small EVs. Moreover, the long-term storage of EVs on CNF sheets was also confirmed, as the CNF sheet could store CD63-positive small EVs at ~10$^{11}$ particles/mL for at least 90 d when the CNF sheet was dry (Supplementary Fig. 4b, c). Furthermore, even more remarkably, the EV concentration recovered from 10 μL of serum using the CNF sheet surpassed that obtained from 250 μL of serum through ultracentrifugation. Additionally, the calculated purity, determined by

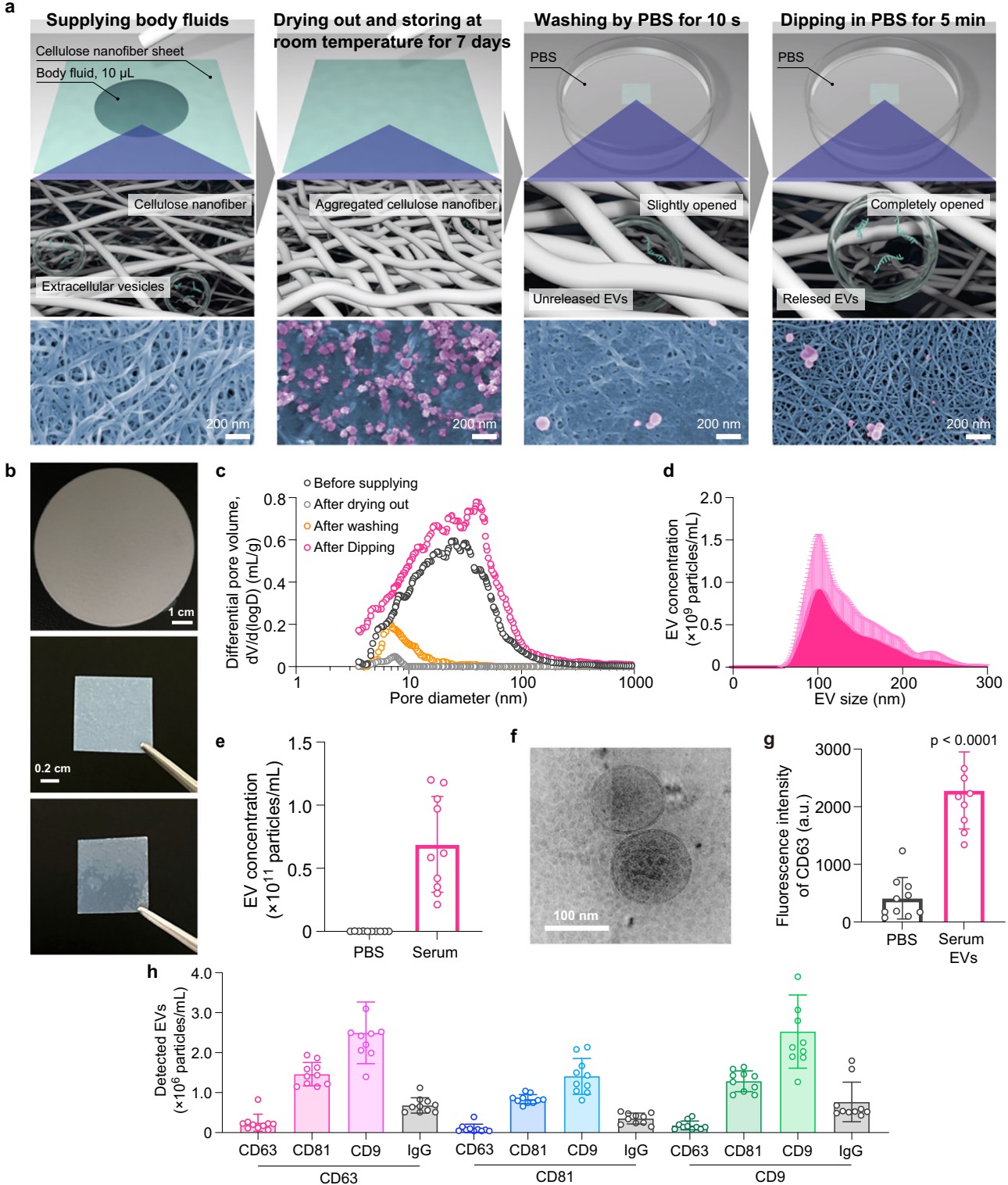

dividing the concentration by the protein content, demonstrated an even higher value with the CNF sheet method (Supplementary Fig. 7). The above proof-of-concept experiment highlighted the significant benefits of the CNF sheet for capturing EVs from wet human tissues, storing them, and even releasing them.

### Small RNA sequencing for EVs in CNF sheets
To further characterize the EVs captured by CNF sheets (CNF sheet EVs), small RNA sequencing (small RNA-seq) was performed by using the total RNAs extracted from CNF sheet EVs of serum (Fig. 2a). In

recent decades, miRNAs in EVs have been well investigated and found to perform essential roles in physiological biology[25,26]. After dropping 10 μL of serum into the CNF sheets and drying and washing the CNF sheets, miRNAs were extracted by immersion in lysis buffer instead of recovery in PBS (Supplementary movie 2), and miRNAs were analyzed by a next-generation sequencing (NGS). The pore size distributions of the CNF sheet after lysis buffer treatment suggested that the lysis buffer could open the pores and release miRNAs inside EVs simultaneously (Fig. 2b). Two independent sequencing analyses were performed on the same sample, and the miRNA profiles were sufficiently

**Fig. 1 | CNF sheets capture intact EVs. a** Conceptual diagram of the CNF operating principle and the corresponding FE-SEM image obtained using a saliva sample. The top row shows an overhead view of the CNF sheet, representing the supply of fluid by pipette, storage and drying for 7 days, washing by immersion in PBS for 10 s, and EVs removal by immersion in PBS for 5 min. The middle row shows a magnified view of CNFs. The bottom row shows SEM images during each experimental process when saliva was used. **b** Photograph of EV sheet (~3 inches in diameter) at the time of production (upper row), before use (middle row) cut into 1 cm squares, and after water absorption and drying (lower row). **c** Pore size distribution before supplying body fluids, after drying, after PBS washing for 10 s, and after PBS immersion for 5 min using the mercury intrusion method. **d** Size distribution of EVs recovered from 10 μL of serum using EV sheets. *n* = 10. **e** Size distribution of contaminants recovered from 0.22 μm-filtered PBS and EVs recovered from 10 μL of serum using

EV sheets. Each dot indicates the respective data value, and error bars indicate SD of a series of measurements (*n* = 10). PBS indicates 10 μL of 0.22 μm-filtered PBS; serum indicates 10 μL of serum from ovarian cancer patients pipetted onto CNF sheets, dried, stored, washed, recovered, and measured by NTA. **f** A CryoEM image of EVs recovered from serum using CNF sheets. **g** Detection of CD63 with fluorescence-labeled antibody using a well plate and a plate reader. CD63 detection of contaminants recovered from 10 μL of 0.22 μm-filtered PBS (denoted as PBS) and EVs recovered from 10 μL of serum (denoted as serum EVs) using CNF sheets. arbitrary units (a.u.) (**h**) Exoview detection of EVs recovered from 10 μL of serum using CNF sheets. CD63, CD9, and CD81 below the line indicate captured antibodies, while CD63, CD9, CD81, and IgG above the line indicate detected antibodies. *n* = 10. **c**–**e**–**g**, **h** Data are presented as mean values with SD. Each experiment was repeated at least three times (**c**–**e**–**g**, **h**).

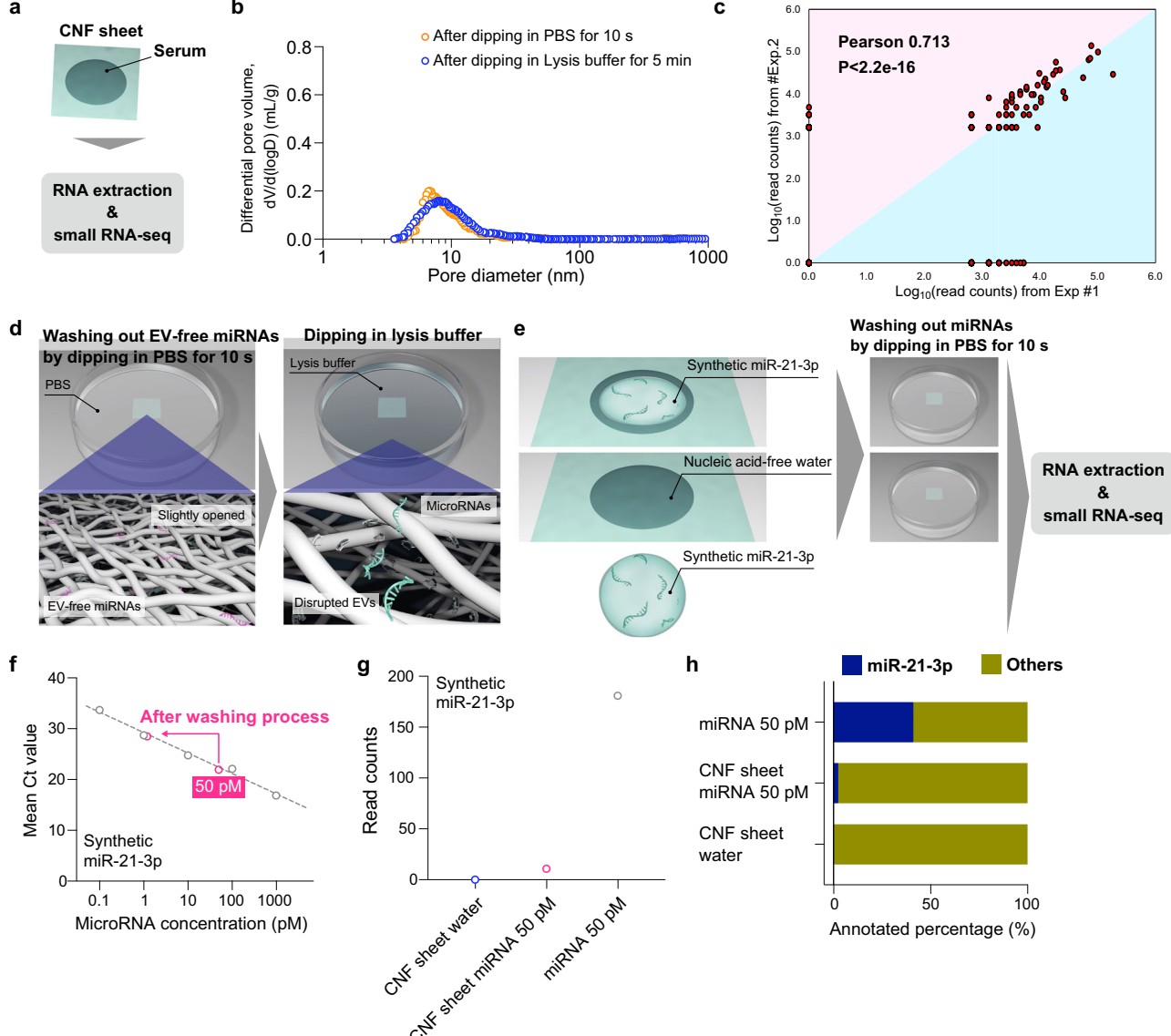

**Fig. 2 | Small RNA sequencing for EVs in CNF sheets. a** Schematic illustrations of RNA extraction and small RNA-seq from serum. **b** Pore size distribution after PBS washing for 10 s and after lysis buffer immersion for 5 min. The experiment was repeated at least three times. **c** Dot plot of miRNA read counts from 2 independent small RNA-seq samples by using the same serum sample. **d** Conceptual illustrations of extracting EV-miRNAs from isolated EVs inside the CNF sheet. **e** Schematic illustrations of examination to check the efficiency of EV-free miRNA

removal. **f** The qRT–PCR results before supplying EV-free miRNAs and after lysis buffer immersion for 5 min. **g** The NGS read count results for the negative control (CNF sheet water), after lysis buffer immersion for 5 min (CNF sheet miRNA 50 pM), and before supplying EV-free miRNAs (miRNA 50 pM). **h** Annotated rate of EV-free miRNAs in the negative control (CNF sheet water), after lysis buffer immersion for 5 min (CNF sheet miRNA 50 pM), and before supplying EV-free miRNAs (miRNA 50 pM).

comparable because of high reproducibility (Fig. 2c). These data suggested that those miRNAs were definitely present in CNF sheet EVs. In addition, 7 days of CNF sheet drying resulted in higher read counts and annotated ratio as miRNAs than 1 day of drying (Supplementary Fig. 8). Considering the number of miRNA reads, EV concentrations, and CD63 expression, the number of storage days was set to 7 days in subsequent experiments.

In our methods of RNA extraction from CNF sheet EVs, EV-free miRNAs were washed out, and then RNAs were collected by lysis buffer (Fig. 2d). To examine the efficiency of EV-free miRNA removal by the washing operation (the 10-s washing with PBS), 10 μL of synthetic miRNA at a concentration of 50 pM was dropped into the CNF sheet, the CNF sheets were dried and stored for 7 days, and after subsequent washing, miRNA extraction using lysis buffer and quantitative evaluation using quantitative reverse transcription PCR (qRT–PCR) and small RNA-seq was performed (Fig. 2e). Considering the number of reported miRNA types and miRNA concentrations in serum, the concentration of EV-free miRNA was set at 50 pM. Quantitative evaluation by qRT–PCR confirmed that 50 pM EV-free miRNA became 1 pM during the washing operation (Fig. 2f). Small RNA-seq also confirmed that the number of reads at 50 pM decreased from 180 to 11 with the wash operation (Fig. 2g). Correspondingly, the annotation rate of EV-free miRNAs decreased from 41% to 2% (Fig. 2h). Note that the other miRNAs here might be due to external contamination or error reads. Taken together, these results suggested that CNF sheets did not capture free miRNAs and that small RNA-seq for CNF sheet EVs enabled the analysis of EV-dependent miRNAs.

## CNF sheet attachment method in vivo
Since the CNF sheets were able to capture EVs from a tiny volume of body fluids, we investigated utilizing this method of EV capturing by CNF sheets to capture a tiny amount of ascites on organs, including the ovary, uterus, peritoneum, omentum, liver, and other organs in the peritoneal cavity. The actual EV capture process from moistened organs using the CNF sheet consists of the following steps: EV capture by absorption of ascites on organs, EV storage by drying, washing, and EV release (Fig. 3a, Supplementary movie 3). Once CNF sheets were attached to the moistened organs, the sheet captured EVs from the surface. After the drying and washing process, intact EVs were isolated. To test this process, CNF sheets were attached to the peritoneal wall of mice that did not have apparent ascites (Fig. 3b). The EVs collected from the surface of peritoneal wall were characterized and confirmed as EVs, with a size distribution of 100–250 nm (Fig. 3c), EV concentration of ~$10^9$ particles/mL (Fig. 3d), spherical shape covered with lipid bilayers (Fig. 3e), and membrane proteins known to be EV markers (Fig. 3f). After confirmation of this CNF sheet attachment method, we hypothesized that EVs at different locations might have different EV profiles. The EVs on the liver surface and peritoneum were collected from mice (Fig. 3g), and small RNA-seq was performed. Although there were certain universal miRNAs, each set of EVs had a unique profile. Five miRNAs, mmu-miR-615-3p, mmu-miR-196b-5p, mmu-miR-196a-5p, mmu-miR-10b-3p and mmu-miR-10b-5p, were highly expressed in EVs from the pelvic peritoneum compared to the liver surface, and in contrast, 12 miRNAs, mmu-miR-335-5p, mmu-miR-214-3p, mmu-miR-199a-5p, mmu-miR-106b-3p, mmu-miR-31-5p, mmu-miR-93-5p, mmu-miR-126a-5p, mmu-miR-126b-3p, mmu-miR-126a-3p, mmu-miR26b-5p, mmu-miR-122-3p and mmu-miR-802-5p, were highly expressed in EVs on the liver surface. Those EVs were conventionally analyzed as a single body fluid, ascites, but this method can independently analyze those EVs with location information (Fig. 3h). These data can shed light on an unveiled aspect of EV biology because miRNAs in EVs are strongly expected to have biological functions, and the location difference can contribute to understanding detailed EV-miRNA mechanisms.

## CNF sheets in ovarian cancer models
Next, to test whether CNF sheet EVs capture physiological EVs, ovarian cancer models were used because cancer cells frequently metastasize into the peritoneal cavity, and the EVs in ascites play an important role[11,27]. ID8 cells, which are mouse ovarian carcinoma cells, were injected intraperitoneally (IP), and tumor growth was monitored by an in vivo imaging system (Fig. 4a). This mouse model accumulates ascites later than 4 weeks, but we sacrificed the mice at day 14. At that time, there were no apparent ascites, but EVs were able to be collected by CNF sheets from tumor-bearing mice or normal mice (Fig. 4b). Small RNA-seq showed that miRNAs in CNF sheet EVs had distinct profiles (Fig. 4c) that closely resembled the cancer tissue profile (Fig. 4d). Compared to normal mice, there were many differentially expressed miRNAs in tissue and CNF sheets (Fig. 4e), and those miRNAs were found by pathway analyses to be associated with cancer-related miRNAs (Fig. 4f). To examine the earlier phase of tumor progression, CNF sheet analyses were performed in orthotopic ovarian cancer mouse models, in which mice were injected with ID8 cancer cells in the ovarian bursa to represent tumor progression from focal diseases (Fig. 4g). The profiles of CNF sheet EVs on day 4 were significantly different from those on day 0 (Fig. 4h), and the profiles migrated towards those of day 28 ascites (Fig. 4i). In summary, the CNF sheet attachment method successfully captured physiological EVs and enabled analysis of the very early phase of tumor progression.

## CNF sheet analyses for ex vivo human tissue
To test the performance of the CNF sheet attachment method in human tissues, EVs on freshly removed tumor tissues were collected (Fig. 5a) from ovarian cancer patient 1 (Supplementary Fig. 9a), and lipid bilayered vesicles were observed (Fig. 5b, Supplementary Fig. 9b). EVs from whole ascites samples (the left panel in Fig. 5a) and those on the tumor surface were compared based on miRNA profiles (Fig. 5c). The clustering analysis revealed that EVs on the tumor surface have unique miRNA profiles relative to tumor tissue and whole ascites. In general, serum and urine are frequently utilized as biofluids for EV analyses. For this reason, we included the serum and urine EVs of the same patient in the PCA analyses. We found that the serum and urine EV profiles were much different from ascites and tumor EV profiles (Fig. 5d). The profile of tumor surface EVs was more similar to tumor tissue profiles than whole ascites EVs (Fig. 5e), which may suggest that CNF sheets directly captured the EVs released from tumors. These results showed that CNF sheets enable the analysis of ascites EVs on the tumor surface (Fig. 5f) and present the concept that location-based ascites could have unique profiles. By conventional methods, we could only compare tumors to ascites EVs, but we can now analyze EVs released from the tumor. Looking at specific miRNAs, some miRNAs were increased only in ascites EVs, and some were increased in EVs on the tumor surface and tissues (Fig. 5g). Regarding the EVs from tumor fluids, we obtained samples from ovarian cancer patient 2 (Supplementary Fig. 9a), and the profiles of the tumor tissues and the EVs on the tumor surface were similar, although the ascites EVs had distinct profiles (Fig. 5h). Interestingly, CNF sheet EVs from inner metastatic tumors had similar profiles to those of primary tumor tissues, but the tissue profiles of primary and metastatic tumors were different (Fig. 5i). Therefore, this approach using CNF sheets unveiled previously unknown profiles of cancer in patients and may lead to a more detailed understanding of cancer biology.

## Intravital CNF sheet analysis reveals location heterogeneity of EVs
To confirm the utility of the CNF sheets and to analyze CNF sheet EVs in patient bodies, we collected the EVs on the organ surfaces by the EV attachment method during surgery. CNF sheets were sterilized with ethylene oxide gas, and sterility was ensured following guidelines from the Japanese Society of Medical Instrumentation (Supplementary

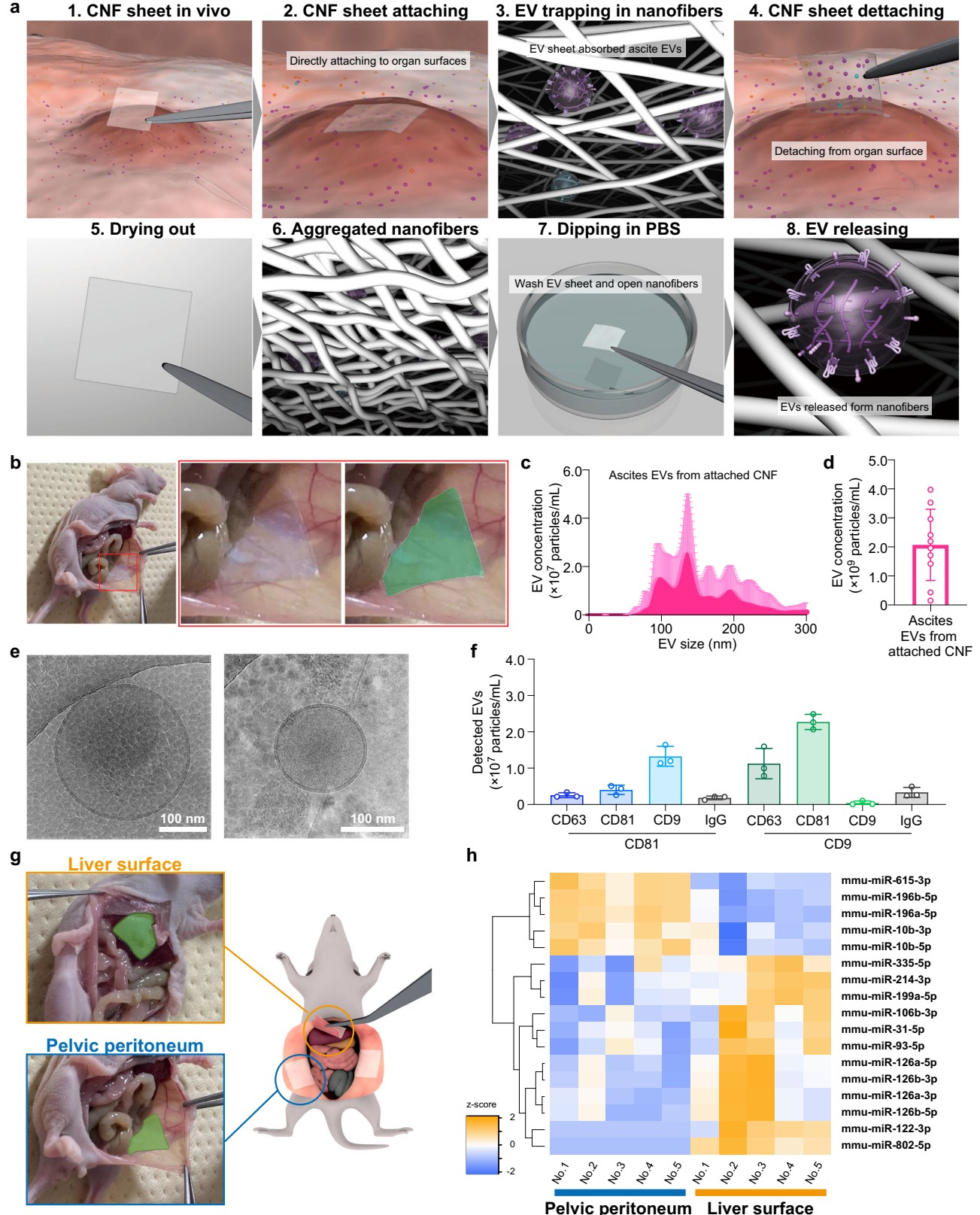

Fig. 10a). After sterilization, we confirmed that the sterilization process had no effect on pore size using FESEM (Supplementary Fig. 11), and we tested the performance of the CNF sheet for EV capture, and it was not affected (Supplementary Fig. 10b). CNF sheet EVs were collected from 4 patients during their abdominal surgeries (Fig. 6a, Supplementary Fig. 10c), and CNF sheet attachment was performed at the pelvic

peritoneum, omentum, liver surface, or tumor surface (Fig. 6b). EV particle concentrations and protein markers were confirmed (Fig. 6c–e, Supplementary Fig. 12). In addition, western blotting analyses revealed that GRP which is the marker protein for cellular compartment was not detected, whereas CD63 was positive (Supplementary Fig. 13). Patients 3–5 were diagnosed as stage I, which

**Fig. 3 | CNF sheet attachment method in vivo. a** Schematic illustrations of the CNF sheet attachment method. CNF sheets were directly attached to organ surfaces, where they absorbed ascites EVs. After detachment from the organ surface, the sheets were completely dried out, and the cellulose nanofibers were aggregated. CNF sheets were washed, and cellulose nanofibers were opened. Finally, the EVs were released from the CNF sheets. **b** Representative photos of CNF sheet attachment to the peroneal wall. **c**, **d** The size distribution and concentration obtained by nanoparticle tracking assays (NTAs) for mouse ascites EVs by using CNF sheets. *n* = 10. **e** Cryo-EM image of EVs recovered from CNF sheet EVs from mouse ascites.

**f** Single-particle quantification was performed using the ExoView platform. *n* = 3 from independent mice. The numbers of detected EVs are displayed in bar charts. **g** Schematic illustrations of CNF sheet attachment sites at the liver surface or peritoneum. **h** A heatmap showing 17 differentially expressed miRNAs from CNF sheet EVs based on small RNA-seq. *n* = 5: EVs from the liver surface and *n* = 5: EVs from the peritoneum. Adjusted *P*-values < 0.05, |log2FC| > 1. **c**, **d**–**f** Data are presented as mean values with SD. Each experiment was repeated at least three times (**c**, **d**–**f**).

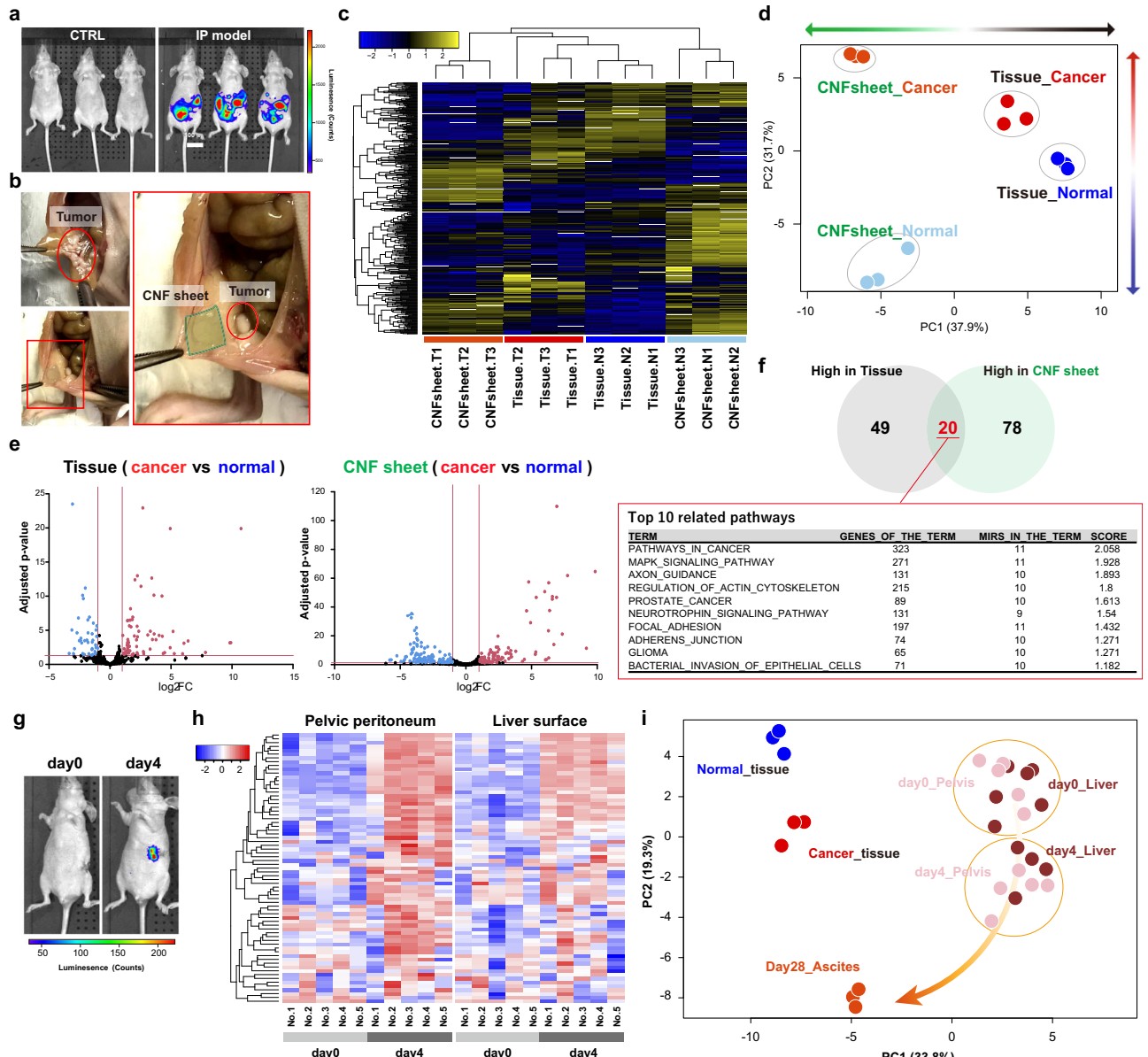

**Fig. 4 | CNF sheet in ovarian cancer mouse models. a** Representative bioluminescence images of the ovarian cancer IP model by using IVIS. *n* = 3. **b** Illustrative photos of tumors, and CNF sheet attachment. **c** A heatmap showing the expression of 485 miRNAs from CNF sheet EVs based on small RNA-seq. CNF sheet T1-3 indicate CNF sheet EVs from tumor-bearing mouse ascites, and CNF sheet N1-3 indicates CNF sheet EVs from tumor-free control mouse ascites. Tissue samples T1-3 were from tumor tissues, and tissue samples N1-3 were from normal ovaries. **d** PCA mapping of miRNA expression from tissue miRNAs and CNF sheet miRNAs from ovarian cancer mouse models. **e** Volcano

plots of differentially expressed miRNAs. Adjusted *P*-values < 0.05, |log2FC| > 1. **f** Venn diagrams show 20 overlapping miRNAs in tumor tissues and CNF sheet EVs in ascites and the results of pathway analysis using 20 miRNAs. **g** Representative bioluminescence images of the ovarian cancer orthotopic mouse model by using IVIS. **h** A heatmap showing differentially expressed miRNAs of CNF sheet EVs from the pelvic peritoneum or liver surface. *n* = 5 in each case, and the time points were days 0 and 4. **i** PCA mapping for miRNA expression by using CNF sheet EVs or tissues. Day 28_Ascites indicated an advanced-stage condition with spontaneous ascites accumulation.

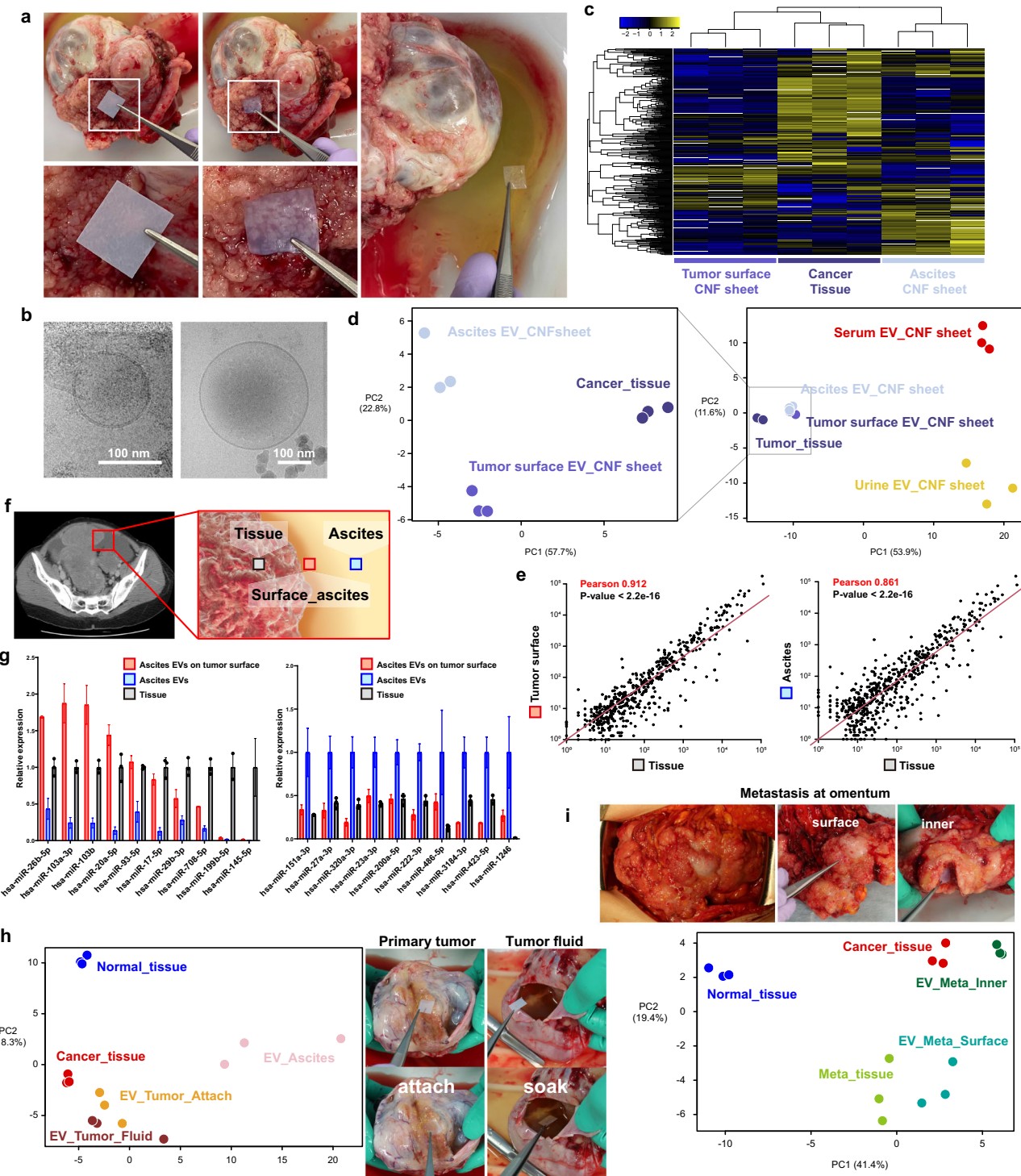

**Fig. 5 | CNF sheet analyses of ex vivo human tissue. a** Illustrative photos of CNF sheets attached to ex vivo human cancer tumors. The right 4 panels indicate CNF sheet attachment, and the left panel indicates CNF sheet soaking for ascites. **b** Cryo-EM image of EVs recovered from CNF sheet EVs on the tumor surface. **c** A heatmap showing differentially expressed miRNAs from CNF sheet EVs or tissues based on small RNA-seq. $n = 3$ each: tumor surface CNF sheet, ascites CNF sheet and cancer tissue. **d** PCA mapping of miRNA expression, $n = 3$ each: tumor surface CNF sheet, ascites CNF sheet cancer tissue, serum_ CNF and urine CNF sheet. **e** Dot plot showing the correlation of miRNA expression in tumor surface CNF sheets vs. cancer tissue and ascites CNF sheets and cancer tissue. **f, g** Schematic illustrations of the location of samples and bar charts represent relative miRNA expression. $n = 3$ from independent locations. Data are presented as mean values with SD. The values in the left graph were normalized to tissue miRNA expression, and those in the right graph were normalized to ascites EVs. $n = 3$ (**h**) PCA mapping and illustrative photos from patient 2 focusing on tumor fluid profiles. **i** PCA mapping and illustrative photos from patient 2 focusing on metastatic tumors in the greater omentum.

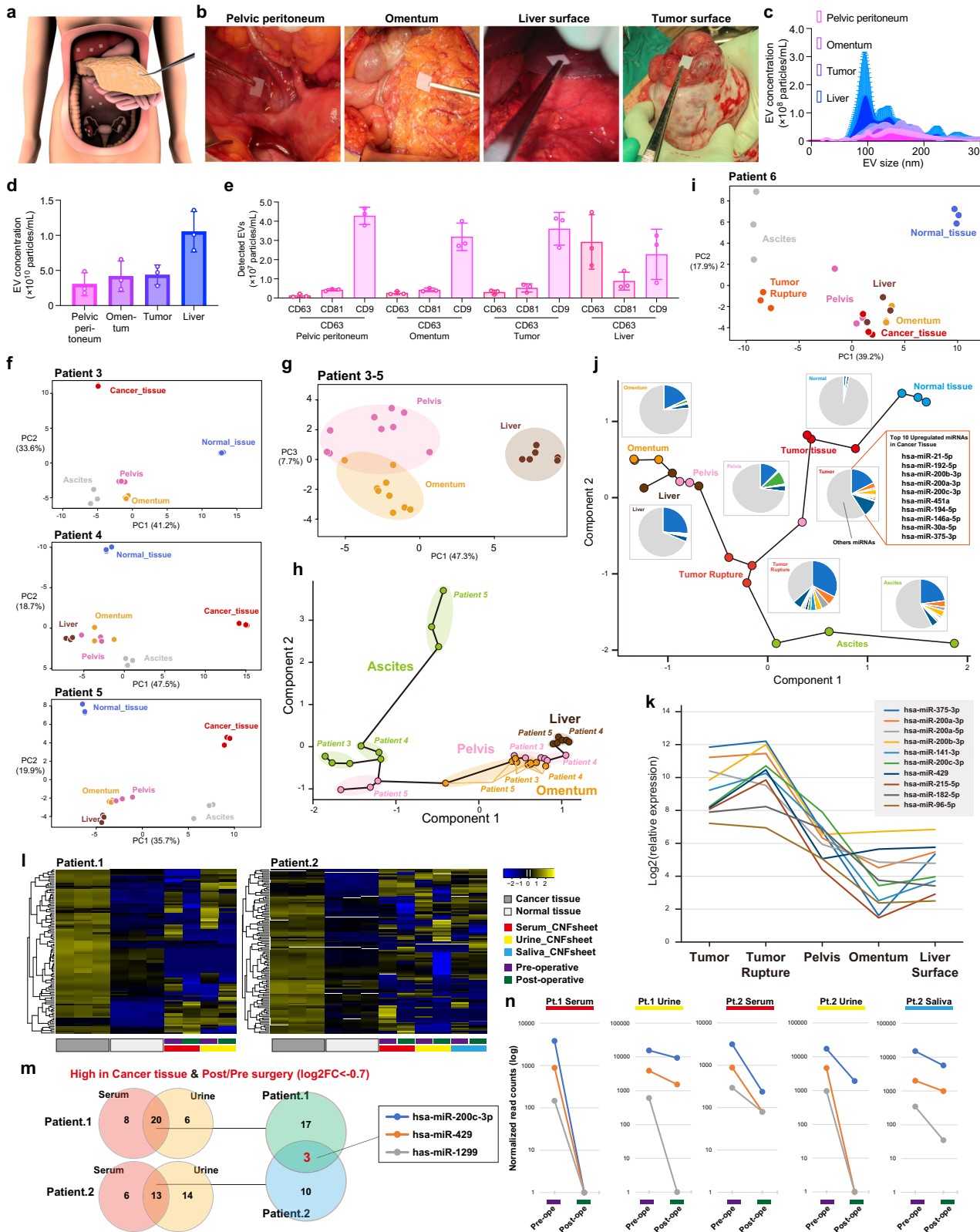

means they had localized disease (Supplementary Fig. 9a), and the miRNA profiles from those patients showed similar patterns (Fig. 6f). In addition, focusing on the location of EVs, the miRNA profiles of each location, including the pelvic peritoneum, omentum, and liver surface, formed clusters (Fig. 6g). To reveal the underlying molecular interaction, a trajectory analysis was performed (Fig. 6h). Ascites EVs were located in a unique area, and all EVs obtained by the attachment

method were connected. In contrast to patients 3–5, patient 6 had progressive disease in that the tumor surface was ruptured and cancer cells had metastasized into the peritoneal cavity (Supplementary Fig. 9a). As in the PCA, the profiles of CNF sheet EVs from all locations were close to those of cancer tissues (Fig. 6i). The trajectory analysis was performed by using these samples and revealed that the pattern of connection was different from patients with localized disease (Fig. 6j).

**Fig. 6 | Intravital CNF sheet analysis reveals location heterogeneity of EVs.**
**a** Schematic illustrations of the locations of CNF sheet attachment during surgery.
**b** Illustrative photos of the locations of CNF sheet attachment, including the pelvic peritoneum, omentum, liver surface and tumor surface. *n* = 3 from independent locations. **c**, **d** The size distributions and concentrations obtained by nanoparticle tracking assays (NTAs) of CNF sheet EVs at the pelvic peritoneum, omentum, liver surface and tumor surface. *n* = 10 (**e**) Single-particle quantification was performed using the ExoView platform. The numbers of detected EVs are displayed in bar charts. **f** PCA mapping of miRNA expression from patients 3–5. Normal tissues indicate the contralateral ovarian tissue without cancer. **g** PCA mapping of miRNA expression in the peritoneum, omentum and liver of patients 3–5. **h** The mapping of trajectory analysis based on miRNA expression in the peritoneum, omentum and liver of patients 3–5. **i** PCA mapping of miRNA expression of patient 6. **j** The mapping of trajectory analysis based on miRNA expression of normal tissue, tumor tissue, CNF sheet EVs on ruptured tumor surface, peritoneum, omentum and liver of patient 6. Pie charts indicate the top 10 upregulated miRNAs in cancer tissues. **k** The line graph indicates the relative miRNA expression of the 10 upregulated miRNAs in cancer tissues in each sample. **l** Heatmaps of miRNA expression from patients 1 and 2, including pre- and postoperative serum-, urine- and saliva-derived CNF sheet EVs. According to the subsequent DEseq analyses, the adjusted *P*-values < 0.05, |log2FC | > 1, 102 miRNAs in patient 1 and 110 miRNAs in patient 2 were differentially expressed in cancer tissues compared to normal tissues. **m** Venn diagrams show the rationale for selecting 3 miRNAs as biomarker candidates. These values were high in cancer tissue, and the expression change from presurgery to postsurgery was defined as log2FC < −0.7. **n** Line graphs showing expression change from presurgery to postsurgery for 3 miRNAs. **c**–**e** Data are presented as mean values with SD. Each experiment was repeated at least three times (**c**–**e**).

---

Some miRNAs were high in tumors and in EVs from tumor rupture sites, and the level of each miRNA decreased with distance from the primary tumors (Fig. 6k). In summary, the location-based heterogeneity of ascites EVs was confirmed, and the profiles indeed reflected disease conditions. Finally, to evaluate the possibility of biomarker applications, pre- and postoperative biofluids, including serum, urine, and saliva, were obtained, and EVs were isolated by CNF sheets. Small RNA-seq was performed, and cancer-related miRNA profiles were compared to normal tissue profiles (Fig. 6l). The miRNAs that were high in serum and urine EVs and high presurvey were assessed, and hsa-miR-200c-3p, hsa-miR-429, and hsa-miR-1299 were selected (Fig. 6m). In addition, these miRNAs decreased post-surgery (Fig. 6n). Regarding further applications, the validation assessment by qRT-PCR is important to provide usability. First, we tested the expression of hsa-miR-200a-3p, hsa-miR-200b-3p, hsa-miR-200c, hsa-miR-148a, hsa-miR-429 and hsa-miR-1299 in the serum EVs which captured by CNF sheets. Among these targets, primers of hsa-miR-200a-3p, hsa-miR-200b-3p worked and those expression decreased post-surgery (Supplementary Fig. 14a, b). Regarding in tumor tissues, hsa-miR-200b-3p and hsa-miR-429 were confirmed as highly expressed in tumor tissues compared to adjacent normal tissues or benign tumors (Supplementary Fig. 14a, b). In addition, the expression of hsa-miR-200b-3p and hsa-miR-429 were high in the EVs from tumor surface ascites than that from whole ascites (Supplementary Fig. 14a, b). Taken together, EV-miRNAs which were obtained by using CNF sheet are promising for biomarker applications.

## Discussion

Although evidence of the functions of EV in disease mechanisms has been accumulating, the EV isolation method always affects the experimental results because different EV isolation methods can isolate different EV subpopulations[28]. EV isolation strategy is still a major challenge in this field and even in EV biology, and technical obstacles to EV isolation can narrow the opportunities for biological understanding[29]. In this study, we showed that CNF sheets enabled the capture of EVs from small amounts of biofluid and confirmed that the subtype of these EVs was CD63-positive small EVs. By using the CNF sheet attachment method which we developed, we found EVs on organ surfaces representing previously unanalyzed EV subpopulations that can now be subjected to biological assessments. EVs on the liver surface or pelvic peritoneum have unique profiles, and several tumor-suppressive miRNAs have been identified in liver surface EVs, such as miR-122-3p[30]. This might be consistent with actual ovarian cancer cell progression, which tends to metastasize to the diaphragm rather than to the liver surface from the space between the diaphragm and liver[31]. Thus, new methodologies can lead to the development of original research focuses and hypotheses.

CNF sheets are fabricated by cellulose nanofibers with tailored porous nanostructures, and this application and optimization were the most important aspects for realizing EV isolation. Cellulose nanofibers were the best materials for building fine nanostructures to supplement small EVs. In this study, we achieved intentional opening/closing/opening of the porous nanostructures of nanopaper by the surrounding liquid conditions. The presence of a low-surface-tension liquid such as *t*-BuOH allowed the researchers to prepare CNF sheets with open porous nanostructures. Based on hydroxyl groups and water on the cellulose nanofiber surface, the CNF sheet automatically changed to densely packed nanostructures (by closure of the porous nanostructures) during the drying operation, acquiring oxygen barrier properties. As with dried food, incorporating water opened the porous nanostructure again. Some contaminants were washed away by a 10-s opening, and the EVs were recovered by a 5-min opening. When biocompatibility, water absorbency, high wet strength (Supplementary Fig. 16), and oxygen barrier properties are taken into account in the intentional opening/closing/opening of the porous nanostructure, it is difficult to find other competing materials. In this study, miRNAs were mainly analyzed to understand molecular profiles, and as shown in Fig. 2, EV-free nucleic acids did not contaminate the CNF sheets. Obtaining high-purity EVs and eliminating nanosized contaminants is always challenging, and the CNF sheets showed desirable performance.

One of the limitations of EV capturing by CNF sheet is the difficulty of proving how specifically the sheet isolates EVs from biological fluids, and the samples can possibly include non-EV contaminations, such as mixed lipoproteins. This issue also exists in all conventional methods, including serial centrifugation, density gradient centrifugation, size-exclusion chromatography, or any other commercially available isolation kits. It remains technically challenging to prove how much lipoproteins are contaminated in the samples and how to exclude them. Due to the uniqueness of the CNF sheet method, comparing it to conventional methods is also challenging. However, we demonstrated that the sheet can isolate EVs with better performance than serial ultracentrifugation, which is a gold standard method. Regarding lipoproteins, many functions of miRNA regarding intracellular transportation by HDL[32] were reported, and LDL can also transport miRNA in body fluids[33]. However, the size range of these dominant two lipoproteins is around 10 nm, and they are too small to be captured in the CNF sheet. In addition, there are several reported miRNAs, including has-miR-223, has-miR-24, has-miR-135*, has-miR-92a, has-miR-486, has-miR-92a, has-miR-146a or has-miR-33, which are known to be associated with LDL/HDL, but these miRNAs were not significantly detected in this study.

Ovarian cancer is one of the deadliest types of cancer, and over 50% of patients die within 5 years of diagnosis[34]. Peritoneal dissemination, which is a severe problem not only in ovarian cancer but also in many intraperitoneal malignancies, is a crucial prognostic factor for patients due to the lack of effective treatment, but the mechanism of its progression is still largely unknown[35]. There is no doubt that ascites is a key factor, and interestingly, cancer cells show organotropism, such as the tendency for ovarian cancer cells to easily metastasize to the greater omentum[36]. This study revealed that the

surface of each organ can have unique EVs, which may have important functions for cancer cells that travel from primary tumors. Further detailed functional analyses of this aspect can reveal the unknown mechanisms, and may contribute to identifying future therapeutic targets. In addition, monitoring EV profiles at multiple sites in the peritoneal cavity may provide a detailed status of disease progression. In ovarian cancer, most cases are diagnosed in the advanced stages, in which the tumor invades beyond the pelvic space. However, disease staging depends largely on the physician's physical evaluation. This non-invasive evaluation of molecular profiles from multiple intra-peritoneal sites may provide clinically-relevant information, such as recurrence risks.

In cancer biology, tumor heterogeneity is a complex concept[37,38]. EV heterogeneity in human body fluids may have two main aspects. The first is heterogeneity due to the molecular differences of each cargo. Some EVs are positive for specific protein markers, but others are not. This difference is now well recognized, and the guidelines from the International Society for Extracellular Vesicles (ISEV) emphasizes understanding this before dealing with EV-related experiments[9]. The second is heterogeneity of composition, which varies from place to place, even in the same body fluid. Organs are composed of diverse cells, and every single cell actively releases EVs[39,40]. Even on an organ-by-organ basis, distinct EVs are expected to be released, and the mixture of these different organ-derived EVs circulates in the body fluid. In this study, ascites fluids were examined and found to be a suitable model for observing such differences, and indeed, location-based EV heterogeneity was proven.

Our ultimate goal is to apply CNF sheets in real clinical practice, and to achieve this, there are several tasks to perform. The first is to develop manufacturing processes to supply large quantities of CNF sheets. At this time, cellulose nanofibers as medical devices do not exist, but now the market is expanding and is ready for mass production. In addition, celluloses are easily available biomaterials and are not toxic to the human body. The second is to define the analysis process. Here, we focused on small RNAs in EVs, but EVs carry diverse bioactive molecules[5,41]. The process will depend on the targets in EVs. One of the strengths of CNF sheets is their high potential for EV preservation, which means that we can use them as a container for EVs. The sheet can be stored in dry conditions for a week and is therefore suitable for transportation. Furthermore, these processes do not alter the characteristics of EVs. The third is to determine actual clinical usage and to establish proofs of concept. Although we suggested the utility as a biomarker in the last part, we have to determine whether we envision a diagnostic tool for cancer screening in healthy people or use during the actual surgery. While any clinical application methods could be beneficial tools, robust validation in a larger sample size is necessary.

In conclusion, we show this EV analysis platform using CNF sheets that can capture EVs with high purity from trace volumes of biofluids and preserve them under dry conditions over a week. CNF sheet attachment methods enable the analysis of EV location heterogeneity in human bodies and can greatly contribute to new medical applications and the elucidation of pathological mechanisms.

## Methods

### Patient samples
Between April 2020 and March 2022, serum and ascites samples of ovarian carcinoma patients were collected at Nagoya University Hospital (Nagoya, Japan), and tissue samples were also obtained under strict ethical approval by the Ethical Committee of Nagoya University Hospital (approval number 2017-0053). For the CNF sheet attachment method, microvolume ascites samples were collected during open abdominal surgeries. In this study, undiluted ascitic fluid was collected (without pelvic washing prior to sampling). All procedures were performed as a clinical intervention trial under strict ethical approval by

the Ethical Committee of Nagoya University Hospital (approval number 2021-0303). Written informed consent was obtained from all patients. Consent to publish clinical information potentially identifying individuals (e.g., age, gender, stage, histological subtype, etc.) was obtained. Serum was centrifuged at 1500 × g for 10 min at room temperature, and undiluted ascites was centrifuged at 300 ×g for 5 min at 4 °C to remove cell debris. The serum and ascites samples were stored at −80 °C until further use.

### In vivo experiments
For in vivo mouse experiments, 8-week-old female BALB/cSlc-nu/nu mice were purchased from Japan SLC, Inc. (Shizuoka, Japan). Mice were cared for in accordance with the Act on Welfare and Management of Animals (Act No. 105 of October 1, 1973) in Japan and the Regulations and Guidelines of Animal Care and Use at Nagoya University. All studies and experiments were supervised and approved by the Center for Animal Research and Education (CARE) at Nagoya University (approval number M220393-004). All mice were housed in an animal facility at Nagoya University School of medicine under the condition of standard 12 h/12 h light/dark cycles with food, water, and diets. The rooms were 18-24 degrees celsius and around 50% humidity. The mice were 8–10 weeks old at the time of tumor intraperitoneal (IP) cell injections or orthotopic injection into the left ovarian bursa (IB)[11]. ID8 luciferase–labeled cells were injected into each mouse (IP: $5 \times 10^6$, and IB: $1 \times 10^6$). An IVIS Spectrum imaging system (Caliper Life Science, Hopkinton, MA) was used to monitor tumor growth. The mice were administered 150 mg kg$^{-1}$ D-luciferin (Promega, Madison, WI) by intraperitoneal injection. The photons in the whole bodies of the animals were measured, and the data were analyzed using LIVINGIMAGE software (Calliper Life Science). Tumor development was monitored every week by IVIS. The ethics committee in our institutional regulates the tumor size, which should be <10% of body weight, and all tumor burden in this study was not exceeded.

### CNF sheet creation
The CNF sheet was fabricated from cellulose nanofibers. Cellulose nanofibers with widths of 22 ± 8 nm were first prepared using never-dried pulp (softwood bleached kraft pulp kindly provided by Shikoku Paper Sales Co., Ltd., Ehime, Japan), according to our previous report[20]. Then, the CNF sheet was fabricated as follows. An aqueous dispersion of cellulose nanofibers (0.2 wt%, 200 mL) was suction-filtered through a membrane filter (H020A090C, hydrophilic poly-tetrafluoroethylene membrane, pore diameter of 0.2 μm, Advantec Toyo Kaisha, Ltd., Tokyo, Japan). Then, 200 mL of *tert*-butyl alcohol (*t*-BuOH, >99.0% purity, Nacalai Tesque, Inc., Kyoto, Japan) was poured into it and gently filtered. The resulting wet sheet was peeled from the filter and dried by hot pressing at 110 °C for 30 min (1.1 MPa) to obtain the CNF sheet.

### Isolation of EVs from CNF sheet surface
observations of the CNF sheets was performed using FE-SEM (SU-8020, Hitachi High-Tech Science Corp., Tokyo, Japan) at an accelerating voltage of 2.0 kV. Platinum sputtering was conducted for the CNF sheets before FE-SEM observation. Pore size distribution curves of the CNF sheets were obtained by a mercury intrusion method using an AutoPoreIV 9520 (Micrometrics Instrument, Corp., Norcross, USA). Prior to the surface observations and pore size distribution analyses, three groups of CNF sheets (after drying, after washing with PBS for 10 s, and after dipping in PBS for 5 min (Fig. 1b)) were immersed in *t*-BuOH at 35 °C for 3 h, recovered, and then dried in an oven at 40 °C for 2 days to maintain their porous nanostructures. Similarly, the pore size distribution of the CNF sheet was also analyzed after washing out EV-free miRNAs by dipping in PBS for 10 s and then further dipping in cell lysis buffer M for 5 min. The water absorbency of the nanopapers was evaluated by measuring the contact angle of water with a DMs-401

contact angle meter (Kyowa Interface Science Co., Ltd., Saitama, Japan).

### EV extraction from body fluid supply using CNF sheets

A volume of 10 μL of the body fluid sample was introduced on a CNF sheet (10 mm × 10 mm). The fluid was evenly spread on the sheet using a micropipette tip. The CNF sheet was stored and air dried at room temperature for 7 days. Subsequently, the dried CNF sheet was immersed in 1 mL of 0.22 μm-filtered PBS (pH 7.2, Invitrogen) in a Petri dish and washed for 10 s. This washing step aimed to remove any excess EVs adhering on the surface of the CNF sheet. Using tweezers, the washed CNF sheet was placed in a 1.5-mL tube (Proteosave SS, Sumitomo Bakelite) filled with 1 mL of 0.22 μm-filtered PBS was used. The tube containing the CNF sheet was then vortexed for 30 s. This step was performed to facilitate the extraction of EVs from the CNF sheet into the PBS solution. After 5 min, the CNF sheet was removed, leaving the EVs in the PBS solution.

### Characterization of EVs extracted from body fluids using CNF sheets

After the EVs were extracted, their concentration and size were analyzed using a nanoparticle tracking analysis (NTA) instrument (NanoSight LM10 HS; Malvern Panalytical, Ltd.). Video data were collected five times, each with a duration of 60 s. The camera level and detection threshold were set to 15 and 5, respectively. NanoSight NTA 3.2 software was used for data analysis.

### Fluorescence detection of CD63 on EV by a plate reader

A volume of 300 μL of EVs extracted with 0.22 μm filtered-PBS were placed in the wells of a 24-well plate. The plate was covered and incubated at room temperature for 20–24 h. After incubation, the solutions were removed from the wells and 300 μL of blocking agent (consisting of 90 mL of 0.22 μm filtered-PBS, 10 mL of Thermo scientific Bloker BSA [10%] in PBS, and 0.5 mL of Tween20 [Funakoshi]) was added. The plates were incubated at room temperature for 1 h. The blocking agent was removed and the wells were washed twice with 300 μL of 0.22 μm filtered-PBS. To each well, 150 μL of 50× diluted anti-CD63 antibody (1 mg/mL, Cosmo Bio) was added and the wells were incubated at room temperature for 1 h. After incubation, each well was washed with 150 μL of 0.22-μm-filtered-PBS. A volume of 150 μL with 50× diluted FITC-labeled anti-IgG (Cosmo Bio Goat anti-mouse IgG antibody) was added to each well. The plates were cover with aluminum to block light and incubated at room temperature for 1 h. The FITC-labeled anti-IgG solution was removed and the wells were washed thrice with 150 μL 0.22-μm-filtered-PBS. Finally, 150 μL of 0.22-μm-filtered-PBS was added and the samples were measured using a plate reader (Tecan, Spark, Ex485 nm/Em 535 nm).

### RNA extraction from body fluids using CNF sheets

The total RNA was extracted by using an RNA purification kit (Norgen Biotek Corp.) following the manufacturer's instructions. CNF sheets were washed by immersion in 0.22 μm-filtered PBS in a Petri dish for 10 s and placed in 1.5 mL of Lysis Buffer A (Norgen Biotek Corp.). In 5-mL tubes (WATSON), 10 μL of β-mercaptoethanol was added to 1 mL of the lysed solution. The mixture was vortexed for 30 s and left to stand at room temperature for 5 min. After 5 min, the CNF sheet was removed. Next, 1.5 mL of 96–100% ethanol was added and the solution was vortexed for 15 s. An aliquot of 650 μL of the solution was added to the Norgen Kit's column and centrifuged at 6,000 x g for 1 min to discard the waste solution. This step was repeated until all of the sample was processed (~5 times). The column was washed with 400 μL of wash solution prepared by mixing 18 mL of Wash Solution A (Norgen Biotek Corp.) and 42 mL of 96–100% ethanol. The column was then centrifuged at 18,000 x g for 1 min, and the waste solution was discarded. This washing step was repeated three times. Dry spinning was

performed at 18,000 x g for 2 min. The column was placed on a new tube, and 50 μL of Elution Solution A (Norgen Biotek Corp.) was added. The column was centrifuged at 400 x g for 2 min and 18,000 x g for 2 min to elute the RNA. The RNA concentration was measured using Qubit (Thermo Fisher).

### Cell lines

ID8 cells, a murine epithelial ovarian cancer cell line established from C57BL/6 murine ovarian surface epithelial cells, were kindly provided by Dr Katherine Roby (University of Kansas Medical Center). For the in vivo experiment, we used a subclone of ID8, ID8-T6, which is highly metastatic in the peritoneal cavity[42]. Before the experiments, short tandem repeat analysis was performed, and the gene profile was identified and verified by the STR analysis as the original ID8 cells. The cells were cultured in Dulbecco's modified Eagle medium (DMEM)-high glucose (Nacalai Tesque) supplemented with 10% heat-inactivated fetal bovine serum (FBS) and 1% antibiotics and were maintained in humidified cell culture incubators at 37 °C and 5% CO2 and have been routinely tested for mycoplasma infection.

### Cryo-EM

The isolated EVs were visualized using a cryo electron microscope (Terabase Inc., Okazaki, Japan) that can generate high-contrast images of the nanostructures of biological materials without staining procedures that may damage the samples. The natural structure of the sample distributed in solution can be observed by preparing the sample using a rapid vitreous ice-embedding method. The detailed condition is provided in the Supplementary Table.

### Small RNA sequencing

The EVs captured in CNF sheets were washed with 0.22 μm-filtered PBS for 10 s, and the total RNA was extracted by using an RNA purification kit (Norgen Biotek Corp.) according to the manufacturer's instructions. To obtain tissue RNA, a gentleMACS Dissociator (Miltenyi Biotec, Bergisch Gladbach, Germany) was used, and tissues were shredded and crushed with QIAzol (Qiagen, Hilden, Germany) in a gentleMACS M tube (Miltenyi Biotec, Bergisch Gladbach, Germany). Then, the RNAs were extracted by using the miRNeasy Plus Mini Kit (QIAGEN, Hilden, Germany). The total RNA concentration of each sample was measured using a Qubit RNA HS Assay Kit (Thermo Fisher Scientific, Waltham, MA). We prepared small RNA libraries using the NEBNext Multiplex Small RNA Library Prep Set for Illumina (New England Biolabs, Ipswich, MA) and added index codes to attribute sequences to each sample. Next, PCR products were purified using the QIAquick PCR Purification Kit (Qiagen) and 6% TBE gel (120 V, 60 min). Furthermore, DNA fragments corresponding to 140–160 bp (the length of small noncoding RNA plus the 3′ and 5′ adapters) were recovered, and the complementary DNA concentration was measured using the Qubit dsDNA HS Assay Kit and a Qubit 2.0 Fluorometer (Life Technologies, Carlsbad, CA). Finally, single-end reads were analyzed on an Illumina MiSeq or NextSeq (Illumina, San Diego, CA). Synthetic miR21-3p was purchased from Koken Co., Ltd., Tokyo, Japan. The full miRNA expression profiles are stored in the Gene Expression Omnibus (GEO) database (GSE216745, GSE216793).

### Detection of CD63, CD9, and CD81 on EVs by Exoview

Exoview was performed following the manufacturer's instruction. First, the required number of chips from the Human Tetraspanin Kit and Mouse tetraspanin Kit were pre-scanned (NanoView Biosciences). The chips were taken out of the refrigerator and kept at room temperature for 15 min before use. The pre-scanned Chip was placed into the center of a 24-well plate using special tweezers. One chip was placed per well, with the Chip ID facing up and ensuring no contact with the wall of the well. The sample was diluted to a concentration of 106–108 particles/mL with incubation solution (10× diluted). A volume

of 50–70 µL of the diluted sample was introduced on the Chip, making sure the pipette tip did not touch the Chip. The 24-well plate was sealed and incubated at room temperature for 16 (±1) hours. After incubation, the Chip was washed with 1000 µL of 10× diluted Solution A by running it along the wall surface to avoid direct contact with the Chip. The Chip was shaken with a special Shaker at 500 rpm for 3 min, repeating this step three times. The antibodies (CD63, CD9, CD81) were diluted using the blocking solution. For example, for a 300 µL adjustment, blocking solution: 300 µL − (0.6 µL × 3) + CD63 0.6 µL + CD9 0.6 µL + CD81 0.6 µL. After the 3rd wash, 750 µL from each well was discarded and 250 µL of the diluted antibody solution was added. The plate was covered with aluminum foil and shaken for 1 h. Next, 500 µL of 10× diluted Solution A was added to each well to make the total volume 1000 µL. A volume of 750 µL from each well was discard and 750 µL of 10× diluted Solution A was added to each well. The plate was shaken again. A volume of 750 µL was discarded, 750 µL of Solution B was added, and the plate was shaken. This was repeated three times. Subsequently 750 µL of Solution B was removed, 750 µL of DI Water (MilliQ is fine) was added, and the plate was shaken. Two 10–15 cm Petri dishes containing Place MilliQ water was prepared for washing the Chip. Removal of the Chip from the 24-well plates was done horizontally to avoid air contact. The Chip was washed in the Petri dish while turning it horizontally about 10 times. The Chip was removed in the same manner and washed in the other Petri dish. The Chip was slowly removed from the Petri dish at an angle of 45°. Finally, the Chip was scanned and analyzed with the analysis software.

### Western blotting analysis

The samples of proteins were loaded onto polyacrylamide gels for electrophoretic separation of proteins at 20 mA. After blocking with Blocking One (Nacalai Tesque Inc., Japan) for 1 h at room temperature, the membranes were incubated overnight at 4 °C with the following primary antibodies: rabbit monoclonal anti-CD63 (EXOAB-CD63A-1; System Biosciences, LLC, CA, USA; dilution 1:1,000) and mouse monoclonal anti-GRP (sc-393402; Santa Cruz Biotechnology; dilution 1:100). The membranes were subsequently washed three times for 5 min each using Tris-buffered saline with 0.1% Tween® 20 (TBST) and incubated for 1–3 h at room temperature with secondary HRP-conjugated mouse anti-rabbit IgG (NA934-1ML; Cytiva Lifesciences, USA; dilution 1:5,000) or anti-mouse IgG (NA931-1ML; Cytiva; dilution 1:2,000) antibodies. The protein ladder is MagicMarkTM XP Western Protein Standard (Thermo Fisher Scientific). The membranes were imaged using ImageQuant LAS 4010 (GE Healthcare, IL, USA). The uncropped blots are shown in Supplementary Fig. 16.

### miRNA expression analyses by qRT-PCR

For each gene, cDNAs were synthesized using the TaqManTM Advanced miRNA cDNA Synthesis Kit (Thermo Fisher Scientific) according to the manufacturer's instructions. TaqMan® Fast Advanced Master Mix (Thermo Fisher Scientific) and TaqManTM Advanced miRNA Assay (Assay ID: 478490_mir for hsa-miR-200a-3p, Assay ID: 477963_mir for hsa-miR-200b-3p, Assay ID: 002300 for hsa-miR-200c, Assay ID: 000470 for hsa-miR-148a, Assay ID: 477849_mir for hsa-miR-429, Assay ID: 478696_mir for hsa-miR-1299 and Assay ID: 001093 for RNU6B; Thermo Fisher Scientific) were used. Then, qPCR was performed using an Mx3000P (Agilent Technologies). The PCR conditions consisted of an initial denaturation step at 95 °C for 10 min, followed by 40 amplification cycles of 95 °C for 15 s and 60 °C for 1 min. The amplified product was monitored by measuring the increase in FAM fluorescence intensity. Each experiment was performed in triplicate and repeated at least three times to ensure reproducibility.

### Bioinformatics analysis

The raw data files of small RNA-seq were analyzed with the CLC Genomics Workbench version 9.5.3 program (Qiagen). After adapter trimming, the data were mapped to the miRbase 22 database, allowing up to two mismatches, and normalized using reads per million mapped reads. Then, RStudio (RStudio, Boston, MA) and R software (ver. 4.0.3) were used. The heatmap.2 function of the gplots package (ver. 3.1.0) was used for the heatmap and hierarchical clustering analysis. To visualize the volcano plots, log2-fold change and adjusted $p$-values for each gene were calculated using the Wald test in DESeq2 (ver. 1.30.0). Trajectory analysis was performed with the 'monocle' package for tracing EVs based on miRNA expression profiles[43]. Dimensionality reduction in the trajectory plot was performed by the 'ICA' method with the expressed miRNAs among the samples.

### Statistical analysis

The unpaired Student's $t$-test for continuous variables was used for comparisons between two groups. The limit of statistical significance for all analyses was defined as a two-sided $P$-value of 0.05.

### Reporting summary

Further information on research design is available in the Nature Portfolio Reporting Summary linked to this article.

## Data availability

All small RNA sequencing data generated in this study have been deposited in the Gene Expression Omnibus database under accession codes "GSE216745", "GSE216793". For miRNA expression, the data were mapped to the miRbase 22 database (https://mirbase.org/). All other data supporting the findings of this study are available within the article and its supplementary files. Any additional requests for information can be directed to, and will be fulfilled by, the corresponding authors. Source data are provided with this paper.

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

## Acknowledgements

The authors wish to acknowledge the Division for Medical Research Engineering, Nagoya University Graduate School of Medicine and to deeply appreciate all patients who agreed to participate in the study. We also thank Dr. Yoshihiro Koya (Nagoya University, Japan) for kindly gifted the ID8 cells. This work was financially supported by the following grants: the Fusion Oriented Research for disruptive Science and Technology (FOREST; JPMJFR204J, to A.Y. and JPMJFR2003, to H.K.), PRESTO (JPMJPR19H9, to T.Y.), and JST SICORP (JPMJSC19E3, to T.Y.) from Japan Science and Technology Agency (JST), a Project for Cancer Research and Therapeutic Evolution (P-PROMOTE) grant number: 22ama221407h0001 (to A.Y.) from the Japan Agency for Medical Research and Development (AMED), Grant Nos. JP21he2302007 (to T.Y.), 22zf0127004s0902 (to T.Y.), and 22zf0127009h0001 (to T.Y.), the New Energy and Industrial Technology Development Organization (NEDO) JPNP20004 (to T.Y.), and JSPS KAKENHI Grant Numbers 21H03075 (to A.Y.), 21H01960 (to A.Y.), 20K21124 (to T.Y.) and 22K18394 (to T.Y.). Moreover, the Princess Takamatsu Cancer Research Fund (to A.Y.), the Daiichi Sankyo Foundation of Life Science (to A.Y.), the Mochida Memorial Foundation for Medical and Pharmaceutical Research (to A.Y.), the Uehara Memorial Foundation (to A.Y.) and the Aichi Cancer Research Foundation (to A.Y.) were also supported.

## Author contributions

A.Y. and T.Y. conceived the project, designed the experiments, and interpreted the results. Each author contributed to this work as follows. A.Y., M.K., Y.N., M.I., S.K., M.Z., and T.Y.:: Isolation of EVs, RNA extraction and library preparation for sequencing. H.Ko., T.Y., and Y.B.: CNF sheet creation and optimization. K.Y., J.N., and Y.Y.: Bioinformatics analysis. A.Y., K.Y., and M.K.: In vivo experiments. A.Y., Y.N., and H.Ka: Patient sample collection. A.Y. and T.Y. wrote the manuscript, and all authors critically reviewed the paper.

## Competing interests

The authors declare no competing interests.

## Additional information

[1]Department of Obstetrics and Gynecology, Nagoya University Graduate School of Medicine, 65 Tsurumai-cho, Showa-ku, Nagoya 466-8550, Japan. [2]Nagoya University Institute for Advanced Research, Furo-cho, Chikusa-ku, Nagoya 464-8603, Japan. [3]Japan Science and Technology Agency (JST), FOREST, 4-1-8 Honcho, Kawaguchi, Saitama 332-0012, Japan. [4]SANKEN (The Institute of Scientific and Industrial Research), Osaka University, 8-1 Mihogaoka, Ibaraki, Osaka 567-0047, Japan. [5]Bell Research Center, Department of Obstetrics and Gynecology Collaborative Research, Nagoya University Graduate School of Medicine, 65 Tsurumai-cho, Showa-ku, Nagoya 466-8550, Japan. [6]Department of Biomolecular Engineering, Graduate School of Engineering, Nagoya University, Furo-cho, Chikusa-ku, Nagoya 464-8603, Japan. [7]Laboratory of Integrative Oncology, National Cancer Center Research Institute, 5-1-1 Tsukiji, Chuo-ku, Tokyo 104-0045, Japan. [8]Department of Oncogenesis and Growth Regulation, Research Institute, Osaka International Cancer Institute, 3-1-69 Otemae, Chuo-ku, Osaka 541-8567, Japan. [9]Institute of Nano-Life-Systems, Institutes of Innovation for Future Society, Nagoya University, Furo-cho, Chikusa-ku, Nagoya 464-8603, Japan. [10]Institute of Quantum Life Science, National Institutes for Quantum Science and Technology (QST), Anagawa 4-9-1, Inage-ku, Chiba 263-8555, Japan. [11]Japan Science and Technology Agency (JST), PRESTO, 4-1-8 Honcho, Kawaguchi, Saitama 332-0012, Japan. [12]Department of Life Science and Technology, Tokyo Institute of Technology, Nagatsuta 4259, Midori-ku, Yokohama 226-8501, Japan. [13]These authors contributed equally: Akira Yokoi, Takao Yasui. ✉e-mail: ayokoi@med.nagoya-u.ac.jp; yasuit@bio.titech.ac.jp

