## [Peer Review File · Nature Communications]

REVIEWER COMMENTS

Reviewer #1 (Remarks to the Author):

This Reviewer focused on small nc-RNAs and miRNAs.

This study demonstrated that the proposed EV-capture system allows the analysis of EV-associated miRNAs. Furthermore, the EV capture system allows the stabilization of EVs and their content (included small RNAs). A result that is of great interest in a diagnostic setting.

The study is methodologically sounding. My only concern is about the use of NGS only for miRNA analysis. NGS, indeed, although rapidly diffusing, still do not represent the elective technology in molecular diagnostics. I suggest to "validate" the results from NGS by qPCR in order to further support the reliability and usability of the findings.

Reviewer #2 (Remarks to the Author):

Ovarian cancer is the most lethal gynecologic cancer and novel approaches are needed to identify mechanisms of metastasis and novel therapeutics. This manuscript presents preliminary data involving a new material designed to capture extracellular vesicles (EVs) through direct contact with moist surfaces, such as internal organs at the time of surgery. My review focuses on the animal studies and potential clinical applications since these are my areas of expertise. The authors should be congratulated for bringing this new technology into gynecologic cancers where there is a tremendous need for novel approaches. Although I can see potentially meaningful applications of this technology from a research standpoint, I have several concerns which limits my enthusiasm. It seems like this manuscript is focused on methods but it lacks experiments to help ovarian cancer researchers see the therapeutic utility of this technology from a therapeutic or diagnostic standpoint.

Major comments

Investigators have not clearly demonstrated that the nanofiber sheet is uncontaminated by the transfer of cells or non-EV cellular material from the surface of organs. This would be important because it would influence the interpretation of all downstream studies. Although I do not doubt the investigators are indeed capturing EVs, I found it difficult to interpret sequencing data which might be contaminated by normal serosal/peritoneal surface mesothelial cells, epithelial cells, cancer cells, or sub cellular debris

from lysis of such cells during the contact and drying process of sheets. Since the sheets are probably creating a nano-scale flow of water from the serosal surfaces to the sheets, it stands to reason that unintentional materials might be captured by the sheets.

In the discussion, the authors suggest that EVs can be used as therapeutic targets but the dissimilarity between EVs obtained at different sites would suggest that a therapeutic strategy to target cancer cells at one anatomic location might fail in another location. This detail is significant because ovarian cancer is a disease that spreads throughout the peritoneum involving multiple organs. A strategy that successfully prevents cancer growth on one organ without addressing growth on other organs would have limited clinical impact. Thus far, the authors have not demonstrated that this strategy of characterizing EVs on multiple organs could someday lead to a meaningful clinical application. In other words, the clinical impact of this technology feels a bit overstated. However, I agree 100% with the authors statement on page 11 that their ultimate goal is to determine the actual clinical usage of this technology (Line 25) because I don't think they are there yet.

Also in the discussion, the authors seem most interested in understanding mechanisms of progression within the peritoneal space. However, ovarian cancer is often diagnosed after progression has already occurred. In other words, understanding the process of dissemination of ovarian cancer has limited clinical/therapeutic relevance after the fact. Prevention of dissemination would presumably require a clinician to identify ovarian cancer before it has spread and presumably therapeutically block engraftment on the surfaces of other organs. Accordingly, it's not easy to see how the authors might develop a therapeutic strategy using this technology for prevention of dissemination/progression. They should expand on this section substantially.

Minor issues made this paper harder to read.

Use of the term "trace ascites" may be controversial with your clinician leaders. This term is typically reserved for visible ascites, such as abnormal densities seen on CT scan or ultrasound, and not typically used in reference to physiologic moisture on the surface of internal organs. The term used on page 7, line 1 "moistened organs" would be more appropriate than "trace ascites". A specific example is on the same page, line 27 which mentions no apparent ascites but the presence of trace ascites. To me, these terms are mutually exclusive.

In some parts, language is a bit informal for a journal of this caliber. For instance, page 7 line 28 refers to sacrifice of laboratory animals but uses casual language: "we took down the mice. . .". I think lay individuals may use terms like this, which is similar to us saying we are 'putting an animal down' but this is too informal for scientific writing.

For all animal experiments, although your reader can count the number of data points in the figures, it would be helpful to also state the number of mice in either the results section or the methods.

Page 8 line 11, readers could probably guess the rationale for testing serum and urine but your readers may find it easier to understand the results if there were clearly stated rationale behind testing these two fluids. It can be disruptive to see this sentence without some context for rationale.

Figure 6. "normal tissue" is not defined.

Page 10 line 13 and 14, the meaning of this sentence is unclear and perhaps the authors did not intend to use the word "fascinate". Also, a hypothesis is not typically elucidated but rather, it should be "tested".

Methods section (page 13, line 13) does not mention whether pelvic washing were done prior to application of nano sheets. It would be standard practice at many institutions to perform pelvic/abdominal washings of the peritoneal surface before resection of the primary tumor or any metastatic sites. This is intended to determine whether malignant cells are loosely present within the peritoneal space. It would be helpful to know whether such washings were performed since it might influence the detection and or distribution of extracellular vesicles and help the reader understand results in figure 6.

Reviewer #3 (Remarks to the Author):

Yokoi et al reported a work on the isolation of extracellular vesicles (EVs) including exosomes using nanocellulose-based paper sheets. The work is very well performed and carefully written. The entire work is highly interesting to researchers working in different of science including materials, biology and medicine. Therefore, I recommend this manuscript to be published in the Nature communication journal after addressing the following issues:

Major comments:

1. Introduction: in this part, the authors did not discuss other EVs capturing substrates such as lipid Patch Microarrays or immune-capturing methods. This should be widely discussed. Though the author discussed the importance of nanocellulose-based matrices for easy capturing EVs, there are other plenty of polysaccharides available. The authors should also discuss this issue in the introduction.

2. Results and discussion: I would suggest to change the term 'extracellular vesicles' sheets since the authors used nanofibrillated cellulose sheets, the name can be changed to either nanocellulose or NFC sheets, something like that, since it creates confusion in numerous places in the manuscript.
3. Swelling capacity of EVs before and after drying should be performed, in all the medium used in this work.
4. I also suggest quantifying the pores and pore volumes in the presence of different medium used in this work and not before and after incubating with PBS. Micro-CT could be used to obtain this data.
5. Please explain the type of interaction between EVs and cellulose molecules
6. The authors did not mention the selectivity of nanocellulose substrates towards EVs and what type of possible components are present from the body fluid used in this work. It appears that any hydrophilic nanofibrous substrates can be used for this work. Therefore, the selectivity of nanocellulose substrates must be ensured and validated; otherwise, the use of nanocellulose for capturing EVs is inappropriate.
7. Page 6, line 38: pls mention what type of organs are discussed here.
8. The morphology of nanocellulose sheets before and after sterilization should be reported; this is not often reported in many other studies. Therefore, it could be interesting for readers in this field.
9. The authors should quantify also the lowest possible capturing limit of nanocellulose sheets.

Reviewer #4 (Remarks to the Author):

In this manuscript, Yokoi et al develop the novel use of cellulose membranes for the capture and downstream analysis of extracellular vesicles from small volumes of tissue and bodily fluids. This concept is novel and could have applications for the use of EVs as biomarkers for many disease states, including cancer as discussed. The use of cellulose membranes for the capture, storage and release is easy in methodology and is therefore a reasonable approach for a clinical setting. While this technology has great potential, I have some reservations about whether the cellulose membranes predominantly capture EVs and therefore question the purity of the preparation during downstream analysis. Whether this is an issue for the overall aim of the technology is a separate problem, it is possible that it does not matter.

The main issue here is that the authors attribute the capture of miRNA specifically to EVs, however this finding is overstated as there are few controls and experimental evidence to support these claims. It is known that miRNA can co-isolate with EVs and lipoproteins. The EV populations isolated are characterised based on size, morphology and surface proteins which is fine, however this does not speak to the purity of the isolated material. The removal of contaminants is discussed, and a 10s step included, however this step is not comprehensively analysed. Therefore, while an interesting

technology, the basic science showing that the cellulose membrane isolates EVs specifically is not strong.

I also find the manuscript difficult to read and analyse because there is insufficient detail in the methods, results and figure legends. There is not enough detail for the work to be reproduced.

Specific comments

Unclear how supp fig 1 supports the sentence beginning on line 5. I assume this is pertaining to water absorbency. It is unclear what supp figure 1 is showing. Please clarify.

Figure 1a – After washing with PBS for only 10s to remove contaminants, it appears that most of the material (EVs?) evident on the surface of the cellulose nanofiber is removed in the SEM image. Also are the SEM images serum EVs? Please update manuscript, methods and figure legend with more information on the SEM. Furthermore, the SEM image of the 5 min wash seem like the fibres are much smaller than the untreated (first image), can the authors comment on this?

Figure 1d - What do the error bars represent? Is this data from one experiment with several reads or several experiments. Which nanoparticle tracking instrument was used; this is not included in the methodology nor is it in a separate NTA specific section. Supp figure 2a - is this sample the 10s wash step or just EV sheets treated with PBS? I see that there is possibly some EVs removed in the 10s wash step by SEM, has this been quantified by NTA. Where is the evidence that contaminants are removed in the wash step?

It is unclear how EV isolation was performed. There is no mention of volumes for each of the wash steps (one to remove contaminants and one to elute EVs) or filtration steps (see next comment) in the methods section. The figure legends are extremely brief. Figure 1e compares 'PBS' and 'Serum' in the figure, again this is not further clarified in the figure legend which states that this shows the size distribution of contaminants recovered from "0.22 um filtered PBS" and "EVs recovered from 10 ul of serum". The same issue in Fig 1g, unclear, and now one of the samples is called 'No EVs'. Is this just 0.22 um filtered PBS treated EV sheets which are then washed 10s then eluted for 5 mins? Please use the same terminology throughout and explain what it means throughout the manuscript (results, figures, methods).

It appears that the 10s PBS wash step after serum treatment is never analysed for EVs or contaminants (lipoproteins/proteins/miRNA) and there is little evidence to support the methodology. Furthermore, there is no analysis for contaminants in the 5 min eluate that contains the EVs. Serum is a complex

biofluid and isolation of EVs without contaminants difficult. This is a major flaw and does not adhere to the MISEV guidelines which the authors are aware.

Little information is given about the use of ExoView. Is this used in Fig 1h and g, if so why are the units plotted differently in the two graphs? The signal is lower for the CD63 antibody across the board (below isotype controls), yet there is comparable signal for CD9 and CD81 when CD63 is used to capture the EVs. This is contradictory and does not support an enrichment in CD63. However, the authors state that the small EVs isolated can be categorised as CD63-positive small EVs, and the majority could be exosomes. However, the EVs are more CD81 and CD9 positive than CD63. There is no evidence that the EVs isolated are exosomes, they are a heterogeneous population of EVs. For examples of enrichment using ExoView see Breitwieser, K.; Koch, L.F.; Tertel, T.; Proestler, E.; Burgers, L.D.; Lipps, C.; Adjaye, J.; Fürst, R.; Giebel, B.; Saul, M.J. Detailed Characterization of Small Extracellular Vesicles from Different Cell Types Based on Tetraspanin Composition by ExoView R100 Platform. *Int. J. Mol. Sci.* 2022, 23, 8544. <https://doi.org/10.3390/ijms23158544>.

Figure 2 figure legend has b and f incorrectly annotated.

The analysis of the removal of free miRNA is not convincing. While it has been shown that EVs contain certain RNAs, it has also been shown that miRNA co-isolate with EVs and lipoproteins. The use of free miRNA as a control does not comprehensively show that the miRNA detected here are in fact inside the EVs isolated on the EV sheets. It would be better to analyse the 10s wash for miRNA after serum treatment. Furthermore, the authors could confirm that the miRNA are 'inside the EVs' by comparing the eluate of the 5 min incubation in lysis buffer with that of PBS.

All the in vivo work is impressive and interesting, however, again it is unclear whether the results can be specifically attributed to EVs. Certainly, the isolates contain EVs but could also contain lipoproteins, protein aggregates, and associated miRNA.

Response to Reviews

Reviewer #1:

This Reviewer focused on small nc-RNAs and miRNAs.

This study demonstrated that the proposed EV-capture system allows the analysis of EV-associated miRNAs. Furthermore, the EV capture system allows the stabilization of EVs and their content (included small RNAs). A result that is of great interest in a diagnostic setting.

The study is methodologically sounding. My only concern is about the use of NGS only for miRNA analysis. NGS, indeed, although rapidly diffusing, still do not represent the elective technology in molecular diagnostics. I suggest to "validate" the results from NGS by qPCR in order to further support the reliability and usability of the findings.

Response: We appreciate your comments, which are helpful in improving our manuscript. As you mentioned, NGS technology is advancing on a daily basis, which has resulted in reduced costs. In more and more cases, validation of transcriptomic data using qRT-PCR is no longer necessary because NGS technologies have become more accessible. Nowadays, numerous companies use NGS as diagnostic tools. However, we agree that other methods for measuring miRNA levels are very important. Therefore, we performed qRT-PCR according to your suggestions.

First, we measured the expression of hsa-miR-200a-3p, hsa-miR-200b-3p, hsa-miR-200c, hsa-miR-148a, hsa-miR-429, and hsa-miR-1299 in the serum EVs captured by CNF sheets using qRT-PCR. Among these targets, hsa-miR-200a-3p and hsa-miR-200b-3p were successfully detected and their expression were decreased post-surgery (Supplementary Fig 14a-b). In malignant tumor tissues, hsa-miR-200b-3p and hsa-miR-429 were confirmed to be highly expressed, at levels greater than in adjacent normal tissue or benign tumors (Supplementary Fig 14a-b). In addition, hsa-miR-200b-3p and hsa-miR-429 levels were elevated in the EVs isolated from the tumor surface than those from whole ascites. Through this analysis, we further demonstrate that EV-miRNAs obtained using CNF sheets are promising for biomarker applications.

We also add the following descriptions in the Method sections.

miRNA expression analyses by qRT-PCR

For each gene, cDNAs were synthesized using the TaqMan™ Advanced miRNA cDNA Synthesis Kit (Thermo Fisher Scientific) according to the manufacturer's instructions. TaqMan® Fast Advanced Master Mix (Thermo Fisher Scientific) and TaqMan™ Advanced miRNA Assay (Assay ID: 478490_mir for hsa-miR-200a-3p, Assay ID: 477963_mir for hsa-miR-200b-3p, Assay ID: 002300 for hsa-miR-200c, Assay ID: 000470 for hsa-miR-148a, Assay ID: 477849_mir for hsa-miR-429, Assay ID: 478696_mir for hsa-miR-1299 and Assay ID: 001093 for RNU6B; Thermo Fisher Scientific) were used. Then, qPCR was performed using an Mx3000P (Agilent Technologies). The PCR conditions consisted of an initial denaturation step at 95 °C for 10 min, followed by 40 amplification cycles of 95 °C for 15 s and 60 °C for 1 min. The amplified product was monitored by measuring the increase in FAM fluorescence intensity. Each experiment was performed in triplicate and

repeated at least three times to ensure reproducibility.

Supplementary Figure 14. qRT-PCR validation for EV-miRNAs. (a) CT values of hsa-miR-200a-3p, hsa-miR-200b-3p, hsa-miR-200c, hsa-miR-148a, hsa-miR-429, and hsa-miR-1299 in the serum EVs captured by CNF sheets. (b) CT values of hsa-miR-200a-3p, hsa-miR-200b-3p, and hsa-miR-429 in the serum EVs captured by CNF sheets. Data from pre- and post-surgery in the same patients were displayed. (c) Relative abundance of hsa-miR-200b-3p and hsa-miR-429 in tumor and normal tissues. Expression levels in tumor tissues were used as the reference and normalized to one. Normal tissues were collected from the contralateral ovary without cancer. (d) Expression of hsa-miR-200b-3p and hsa-miR-429 in malignant tumor tissues from patients 1 and 2 and in a benign tumor confirmed to be cyst adenoma. (e) CT values of hsa-miR-200b-3p and hsa-miR-429 in ascites EVs from patient 2.

Reviewer #2 :

Ovarian cancer is the most lethal gynecologic cancer and novel approaches are needed to identify mechanisms of metastasis and novel therapeutics. This manuscript presents preliminary data involving a new material designed to capture extracellular vesicles (EVs) through direct contact with moist surfaces, such as internal organs at the time of surgery. My review focuses on the animal studies and potential clinical applications since these are my areas of expertise. The authors should be congratulated for bringing this new technology into gynecologic cancers where there is a tremendous need for novel approaches. Although I can see potentially meaningful applications of this technology from a research standpoint, I have several concerns which limits my enthusiasm. It seems like this manuscript is focused on methods but it lacks experiments to help ovarian cancer researchers see the therapeutic utility of this technology from a therapeutic or diagnostic standpoint.

Response: We appreciate your comments, which have helped us improve the manuscript. All points that you raised in this section have been addressed in the revision.

Major comments

Comment #2-1: Investigators have not clearly demonstrated that the nanofiber sheet is uncontaminated by the transfer of cells or non-EV cellular material from the surface of organs. This would be important because it would influence the interpretation of all downstream studies. Although I do not doubt the investigators are indeed capturing EVs, I found it difficult to interpret sequencing data which might be contaminated by normal serosal/peritoneal surface mesothelial cells, epithelial cells, cancer cells, or sub cellular debris from lysis of such cells during the contact and drying process of sheets. Since the sheets are probably creating a nano-scale flow of water from the serosal surfaces to the sheets, it stands to reason that unintentional materials might be captured by the sheets.

Response: Thank you for your comments. We agree that these points are very critical. Theoretically, it would impossible for the sheets to capture cells because their pore size is too small to trap cells. However, it has not been fully proven that non-EV structures can contaminate the sheets. Moreover, other conventional EV isolation methods, including serial centrifugation, size-exclusion chromatography, and filter-membrane kits, can be at risk for such contamination. In this revised manuscript, we compared the purity of EVs obtained using ultracentrifugation and our CNF sheet. The EV purity is higher with our CNF sheet method than with ultracentrifugation. While the specific separation of lipoproteins and protein aggregates mixed with EVs remains a challenge that needs to be addressed in future research, it is evident that this method can isolate a larger quantity of EVs compared to ultracentrifugation.

Supplementary Figure 7. Comparison of concentration and purity of EVs extracted using the various methods. (a) Concentration of EVs extracted from 10 μ L of 0.22 μ m-filtered PBS using the CNF sheet, from 250 μ L of serum using ultracentrifugation, and from 10 μ L of serum using the CNF sheet. (b) EV purity was calculated as the EV concentration (particles/mL) divided by the protein concentration (μ g/mL). EV concentration was measured by NTA and protein concentration was measured by Qubit.

By using conventional EV isolation methods, it is very challenging to obtain perfectly pure EVs without other contaminants. At the least, our method can isolate EVs with better performance compared to serial ultracentrifugation. Regarding lipoproteins, many functions of miRNA regarding intracellular transportation by HDL⁴ were reported, and LDL can also transport miRNA in body fluids⁵. However, the size range of these two dominant lipoproteins is around 10 nm, and they are too small to be captured in the CNF sheet. In addition, there are several reported miRNAs, including has-miR-223, has-miR-24, has-miR-135*, has-miR-92a, has-miR-486, has-miR-92a, has-miR-146a, or has-miR-33, which are known to be associated with LDL/HDL. However, these miRNAs were not detected in this study. Furthermore, it is essential to note that there are no direct results indicating complete removal of lipoproteins. Therefore, we have added a discussion that includes the possibility of mixed lipoproteins in the revised manuscript; **One of the limitations of EV capturing by CNF sheet is the difficulty of proving how specifically the sheet isolates EVs from biological fluids, and the samples can possibly include non-EV contaminations, such as mixed lipoproteins. This issues also exist in all conventional methods, including serial centrifugation, density gradient centrifugation, size-exclusion chromatography, or any other commercially available isolation kits. It remains technically challenging to prove how much lipoproteins are contaminated in the samples and how to exclude them. Due to the uniqueness of the CNF sheet method, comparing it to conventional methods is also challenging. However, we demonstrated that the sheet can isolate EVs with better performance than serial ultracentrifugation, which is a gold standard method. Regarding lipoproteins, many functions of miRNA regarding intracellular transportation by HDL³² were reported, and LDL can also transport miRNA in body fluids³³. However, the size range of these two dominant lipoproteins is around 10 nm, and they are too small to be captured in the CNF sheet. In addition, there are several reported miRNAs, including has-miR-223, has-miR-24, has-miR-135*, has-miR-92a, has-miR-486, has-miR-92a, has-miR-146a, and has-miR-33, which are known to be associated with LDL/HDL. However, these miRNAs were not detected in this study.**

Regarding the CNF sheet method, we agree that potential contamination with cell debris might occur, even if the sheets attached to the organs for just a few seconds. To address this concern, we performed western blotting analysis to check for contamination with cell-derived materials using GRP, a well-known protein marker for the cellular compartment. In EV samples from patients' organ surfaces, GRP was not detected whereas CD63, a well-known EV marker, was detected. Although we understand that this approach may have limitations, these data suggest that contamination of the sheets with cellular debris is minimal. Furthermore, if the sheets only capture the materials from serosal/peritoneal surface cells, location-based differences in the miRNA profiles should not have been observed. In the revised manuscript, we include a discussion on the possibility of contamination with non-EV materials.

Supplementary Figure 13. Western blotting analyses for EV and non-EV protein markers. “Fibroblasts” refer to the HFF2T cells, which are human foreskin derived fibroblasts immortalized with TERT protein markers. “Patient serum” refers to the serum EVs from patient no. 1 (Supplementary Fig. 9) isolated by serial ultracentrifugation. TCL: total cell lysate.

We also add the following sentences in the Result and Method sections.

“In addition, western blotting analyses revealed that GRP, a for the cellular compartment, was not detected, while CD63, an EV marker, was positive”

Western blotting analysis

The samples of proteins were loaded onto polyacrylamide gels for electrophoretic separation of proteins at 20 mA. After blocking with Blocking One (Nacalai Tesque Inc., Japan) for 1 h at room temperature, the membranes were incubated overnight at 4 °C with the following primary antibodies: rabbit monoclonal anti-CD63 (EXOAB-CD63A-1; System Biosciences, LLC, CA, USA; dilution 1:1,000) and mouse monoclonal anti-GRP (sc-393402; Santa Cruz Biotechnology; dilution 1:100). The membranes were subsequently washed three times for 5 min each using Tris-buffered saline with 0.1% Tween® 20 (TBST) and incubated for 1–3 h at room

temperature with secondary HRP-conjugated mouse anti-rabbit IgG (NA934-1ML; Cytiva Lifesciences, USA; dilution 1:5,000) or anti-mouse IgG (NA931-1ML; Cytiva; dilution 1:2,000) antibodies. The protein ladder is MagicMark™ XP Western Protein Standard (Thermo Fisher Scientific). The membranes were imaged using ImageQuant LAS 4010 (GE Healthcare, IL, USA). The uncropped blots are shown in **Supplementary Fig. 14**.

Comment #2-2: In the discussion, the authors suggest that EVs can be used as therapeutic targets but the dissimilarity between EVs obtained at different sites would suggest that a therapeutic strategy to target cancer cells at one anatomic location might fail in another location. This detail is significant because ovarian cancer is a disease that spreads throughout the peritoneum involving multiple organs. A strategy that successfully prevents cancer growth on one organ without addressing growth on other organs would have limited clinical impact. Thus far, the authors have not demonstrated that this strategy of characterizing EVs on multiple organs could someday lead to a meaningful clinical application. In other words, the clinical impact of this technology feels a bit overstated. However, I agree 100% with the authors statement on page 11 that their ultimate goal is to determine the actual clinical usage of this technology (Line 25) because I don't think they are there yet. Also in the discussion, the authors seem most interested in understanding mechanisms of progression within the peritoneal space. However, ovarian cancer is often diagnosed after progression has already occurred. In other words, understanding the process of dissemination of ovarian cancer has limited clinical/therapeutic relevance after the fact. Prevention of dissemination would presumably require a clinician to identify ovarian cancer before it has spread and presumably therapeutically block engraftment on the surfaces of other organs. Accordingly, it's not easy to see how the authors might develop a therapeutic strategy using this technology for prevention of dissemination/progression. They should expand on this section substantially.

Response: Thank you very much for your important comments. We sincerely acknowledge that we have overstated the clinical implications of our work in some parts of the manuscript. Moreover, we acknowledge how difficult it is to overcome ovarian cancer clinically. As you have kindly stated, the ultimate goal of our research is clinical translation. However, our technology is still undergoing evaluation in a study with a larger sample size. Although it will take time to demonstrate efficacy, at least in this study, we intended to show a new technology for capturing EVs and their potential to contribute to understanding the biology of EVs.

Although the information obtained from organ surface EVs will not be used to identify direct therapeutic targets, our data may elucidate new mechanisms that could explain why ovarian cancer cells have metastatic organotropisms. As mentioned in the Discussion section, EVs on the liver surface or pelvic peritoneum have unique profiles, and several tumor-suppressing miRNAs have been identified in liver surface EVs. These observations might be consistent with actual ovarian cancer cell progression, which tends to metastasize to the diaphragm rather than to the liver surface (via the space between the diaphragm and liver). We are now conducting functional studies based on these insights. However, we sincerely understand that preventing cancer dissemination remains very challenging. What we can do now is to elucidate the mechanisms of tumor progression and move the field forward closer to the goal of metastasis prevention.

In terms of clinical impacts, the sheets we developed may be utilized as a platform for biomarker discovery. We

are passionate in using our technique to providing new insights on the role of EVs in ovarian cancer and would like to further explore this technology.

The sentences were revised in the manuscript as following; Further detailed functional analyses of this aspect can reveal unknown mechanisms, and may contribute to identifying future therapeutic targets. In addition, monitoring EV profiles at multiple sites in the peritoneal cavity may provide a detailed status of disease progression. In ovarian cancer, most cases are diagnosed in the advanced stages, in which the tumor invades beyond the pelvic space. However, disease staging depends largely on the physician's physical evaluation. This non-invasive evaluation of molecular profiles from multiple intra-peritoneal sites may provide clinically-relevant information, such as recurrence risks.

Minor issues made this paper harder to read.

Comment #2-3: Use of the term “trace ascites” may be controversial with your clinician leaders. This term is typically reserved for visible ascites, such as abnormal densities seen on CT scan or ultrasound, and not typically used in reference to physiologic moisture on the surface of internal organs. The term used on page 7, line 1 “moistened organs” would be more appropriate than “trace ascites”. A specific example is on the same page, line 27 which mentions no apparent ascites but the presence of trace ascites. To me, these terms are mutually exclusive.

Response:

Thank you very much, and it is very informative for us. We carefully check the parts which the term, trace ascites, was used and removed all of them from the manuscript.

Comment #2-4: In some parts, language is a bit informal for a journal of this caliber. For instance, page 7 line 28 refers to sacrifice of laboratory animals but uses casual language: “we took down the mice. . . “. I think lay individuals may use terms like this, which is similar to us saying we are ‘putting an animal down’ but this is too informal for scientific writing.

Response:

Thank you for pointing this out. This comment is very helpful. Our manuscript underwent English/scientific proof editing by Springer Nature and the point you raised was not previously identified. We carefully reviewed our manuscript and resent it for editing to another editing service provider.

Comment #2-5: For all animal experiments, although your reader can count the number of data points in the figures, it would be helpful to also state the number of mice in either the results section or the methods.

Response: We added the exact number of mice in the revised manuscript.

Comment #2-6: Page 8 line 11, readers could probably guess the rationale for testing serum and urine but your

readers may find it easier to understand the results if there were clearly stated rationale behind testing these two fluids. It can be disruptive to see this sentence without some context for rationale.

Response: We carefully checked the manuscript and revised the following sentences to avoid ambiguity: **In general, serum and urine are frequently utilized as biofluids for EV analyses. For this reason, we included the serum and urine EVs of the same patient in the PCA analyses. We found that** the serum and urine EV profiles were much different from ascites and tumor EV profiles (**Fig. 5d**).

Comment #2-7: Figure 6. “normal tissue” is not defined.

Response: Thank you for your comment. We used normal ovarian tissues when referring to “normal tissue” in the manuscript. We understand that the origin of ovarian carcinomas is diverse, but we used this definition of normal tissue in this study. Additional descriptions were added in the revised manuscript; **“Normal tissues indicate the contralateral ovarian tissue without cancer.”**

Comment #2-8: Page 10 line 13 and 14, the meaning of this sentence is unclear and perhaps the authors did not intend to use the word “fascinate”. Also, a hypothesis is not typically elucidated but rather, it should be “tested”.

Response: Thank you for your comment. This part was revised according to your suggestions.

New methodologies can **lead to** the development of new research questions and new hypotheses to **test**.

Comment #2-9: Methods section (page 13, line 13) does not mention whether pelvic washing were done prior to application of nano sheets. It would be standard practice at many institutions to perform pelvic/abdominal washings of the peritoneal surface before resection of the primary tumor or any metastatic sites. This is intended to determine whether malignant cells are loosely present within the peritoneal space. It would be helpful to know whether such washings were performed since it might influence the detection and or distribution of extracellular vesicles and help the reader understand results in figure 6.

Response: We totally agree that the procedure can significantly affect the results. In all cases, pelvic washing was not performed before sampling. This description was included in Methods section of the revised manuscript. In this study, **undiluted ascitic fluid was collected (without pelvic washing prior to sampling)**.

Finally, we again thank the reviewer for the very constructive suggestions. We hope that the present submitted manuscript is now satisfactory and answers all the reviewer’s concerns.

Reviewer #3 :

Yokoi et al reported a work on the isolation of extracellular vehicles (EVs) including exosomes using nanocellulose-based paper sheets. The work is very well performed and carefully written. The entire work is highly interesting to researchers working in different of science including materials, biology and medicine. Therefore, I recommend this manuscript to be published in the Nature communication journal after addressing the following issues:

Response: First of all, we would like to thank Reviewer #3 for the very constructive suggestions. We would like to address the suggestions as follows.

Major comments:

Comment #3-1. Introduction: in this part, the authors did not discuss other EVs capturing substrates such as lipid Patch Microarrays or immune-capturing methods. This should be widely discussed. Though the author discussed the importance of nanocellulose-based matrices for easy capturing EVs, there are other plenty of polysaccharides available. The authors should also discuss this issue in the introduction.

Response: We understand that there are several methods in the literature to obtain EVs from body fluids, such as serial centrifugation, density gradient centrifugation, and size-exclusion chromatography. Moreover, target-specific technology are available for EV analysis if a target molecule of interest has been pre-determined. In this study, the goal was to implant nanofiber network-derived porous nanostructures in the body and recover EVs. To achieve this, we utilized cellulose because it is the polysaccharide backbone of sterilized gauze, a material with a proven track record of internal attachment. While there are various types of polysaccharides available, only a subset of these can be used to manufacture nanofibers. Among these, cellulose nanofibers have been extensively researched and are gradually entering commercialization. We elaborated on our choice of cellulose in the Introduction section: “Recently, several methods have been developed to obtain EVs from body fluids, such as serial centrifugation, density gradient centrifugation, and size-exclusion chromatography. Moreover, target-specific technologies to analyze EVs, such as lipid patch microarrays or immune-capturing, are applicable if the target molecule has been pre-determined. Cellulose was chosen for our CNFs because it shares the same polysaccharide backbone as sterilized gauze, a material that has been successfully used in technologies applied inside the body. Cellulose nanofibers derived from wood cell walls have attracted considerable attention due to their fascinating properties, such as low density (1.6 g/cm³), high strength (2–3 GPa), large specific surface area (up to ca. 800 m²/g), abundant hydroxyl groups, strong water absorption, sustainability, biocompatibility, and biodegradability¹⁻³. Additionally, cellulose nanofibers can now be produced on a large-scale in industrial operations, with its market projected to reach USD 2.0 billion by 2030.”

While we focus on cellulose in this study, it is worth noting that chitin and chitosan are other polysaccharide nanofibers currently under investigation. If their efficacy and safety can be demonstrated, we plan to include these materials in our future research and publications.

Comment #3-2. Results and discussion: I would suggest to change the term ‘extracellular vesicles’ sheets since

the authors used nanofibrillated cellulose sheets, the name can be changed to either nanocellulose or NFC sheets, something like that, since it creates confusion in numerous places in the manuscript.

Response: According to your suggestion, we revised the nomenclature by replacing “EV sheet” with “cellulose nanofiber (CNF) sheet.” in the entire manuscript.

Comment #3-3. Swelling capacity of EVs before and after drying should be performed, in all the medium used in this work.

Response: We included cryo-EM images (**Supplementary Fig. 6**) of EVs derived from serum and ascites before and after CNF sheet collection. Based on these data, we concluded that the size of EVs remains unchanged.

Supplementary Figure 6. cryo-EM images of serum and ascites EVs obtained using serial centrifugation. Scale bars: 100 nm.

Comment #3-4. I also suggest quantifying the pores and pore volumes in the presence of different medium used in this work and not before and after incubating with PBS. Micro-CT could be used to obtain this data.

Response: We utilized the mercury intrusion porosimetry to measure the pore size and pore volume of CNF sheets after infiltration with and drying of various body fluids, including serum, urine, ascitic fluid, and saliva (**Supplementary Fig. 2**). After drying, we observed that the pore distribution in the size range of 30–200 nm decreased, with more pores beyond this range. Due to the wider range of pore diameters in the material (ranging from 4–300 nm in diameter), micro-CT analysis was unable to measure all the pore diameters accurately. leading us to choose the mercury injection method for this analysis.

Supplementary Figure 2. Pore size distribution in CNF sheets determined using mercury intrusion method after supplying fluid (10 μ L of medium, urine, serum, ascitic fluid, and saliva) and 7 days of drying.

Comment #3-5. Please explain the type of interaction between EVs and cellulose molecules

Response: We propose that EVs and cellulose interact through hydrogen bonding. In a separate manuscript (currently under review), we demonstrated through molecular dynamics simulations and infrared (IR) spectroscopy that oxide nanowires and lipid nanoparticles interact via hydrogen bonds. In aqueous solution, solidified water molecules on the surface of oxide nanowires form hydrogen bonds with the phosphate groups of the lipid nanoparticles. In our study, we posit that the hydroxyl ($-OH$) groups of cellulose form hydrogen bonds with the phosphate groups of the EV lipid bilayer. To confirm the presence of hydrogen bonding between CNF sheets and EVs, we conducted IR spectroscopy (**Supplementary Fig. 3**) of freeze-dried lipid nanoparticles, lipid nanoparticles dispersed in ultrapure water, and lipid nanoparticles on CNF sheet. In the IR spectra of the freeze-dried lipid nanoparticles, phosphate symmetric stretching vibration peaks appeared, indicating intermolecular association in the solid phase (1047 cm^{-1}). When the lipid nanoparticles were redispersed in ultrapure water, peaks related to hydration were observed (1051 cm^{-1}). Upon drying of lipid nanoparticles dispersed in ultrapure water on the CNF sheet, the IR peak position shifted (1060 cm^{-1}), suggesting hydrogen bonding between the phosphate group of lipid nanoparticles and the $-OH$ groups of the CNF sheet.

Supplementary Figure 3. Infrared (IR) spectra of freeze-dried lipid nanoparticles, lipid nanoparticles dispersed in ultrapure water, and lipid nanoparticles on the CNF sheet. The vertical dashed line indicates the wavenumber of 1047 cm^{-1} .

Comment #3-6. The authors did not mention the selectivity of nanocellulose substrates towards EVs and what type of possible components are present from the body fluid used in this work. It appears that any hydrophilic nanofibrous substrates can be used for this work. Therefore, the selectivity of nanocellulose substrates must be ensured and validated; otherwise, the use of nanocellulose for capturing EVs is inappropriate.

Response: As mentioned earlier, cellulose substrates can theoretically capture not only EVs but also proteins, nucleic acids, and metabolites through hydrogen bonding via -OH groups. To ensure efficient capture and storage of EVs while allowing smaller molecules to flow out, we performed washing using a phosphate buffer solution rich in phosphate groups, which have been demonstrated to exhibit strong adsorption ability. By controlling the washing time, we were able to regulate pore size opening (up to 30 nm), allowing relatively large EVs (30-200 nm in diameter) to remain in the pores while allowing smaller molecules, such as proteins, nucleic acids, and metabolites (below 30 nm), to be flushed out.

It can be argued that hydrophilic nanofiber substrates, owing to the importance of hydrogen bonding by -OH groups, is a suitable alternative for this purpose. However, the primary objective of this study is to capture, store, and recover spatiotemporal EVs from minute amounts of body fluids while ensuring biocompatibility. We instead chose cellulose because it shares the same composition as gauze, a material with a track record of biocompatibility and safety. This approach is novel and is reported for the first time. While hydrophilic nanofiber base materials approved for use in the body could potentially serve the same purpose, the challenge lies in processing the material into a sheet while maintaining a nanopore size of 4–300 nm to achieve EV recovery. Obtaining ultrafine nanofibers, such as the $22 \pm 8\text{ nm}$ used in this study, is already a feat in the context of existing technologies. Furthermore, the material is required to possess chemical stability against $t\text{-BuOH}$ treatment,

which is crucial for nanopore structure formation. Considering the current practical usage of cellulose nanofibers worldwide, we believe this material is the ideal choice for our study.

Comment #3-7. Page 6, line 38: pls mention what type of organs are discussed here.

Response: Thank you for pointing this out, and we acknowledge that we should have added more information. We intended to mean here various organs, including the ovary, uterus, peritoneum, omentum, liver, and any organ in the peritoneal cavity. In this study, we focused on ovarian cancer, and therefore, the ovary, uterus, peritoneum, and omentum are considered important targets for investigation. In the revised manuscript, we added the description of target organs; “a tiny amount of ascites on organs, including the ovary, uterus, peritoneum, omentum, liver, and other organs in the peritoneal cavity.”

Comment #3-8. The morphology of nanocellulose sheets before and after sterilization should be reported; this is not often reported in many other studies. Therefore, it could be interesting for readers in this field.

Response: We included an FE-SEM image of post-sterilization CNF sheets in the manuscript (**Supplementary Fig. 11**).

Supplementary Figure 11. FE-SEM image of post-sterilization CNF sheets.

Comment #3-9. The authors should quantify also the lowest possible capturing limit of nanocellulose sheets.

Response: In the case of cell line-derived EVs, we extracted 10^9 particles (1 mL of 10^{11} particles/mL) when the input was 10^9 particles (10 μ L of 10^{11} particles/mL). Similarly, 10^8 particles were extracted from an input of 10^8 particles. However, when 10^{10} particles were added, only about 5×10^9 particles were extracted (**Supplementary Fig. 5**). These indicate that a higher EV input poses more significant challenges than a lower EV input and that 5×10^9 particles represent the maximum amount of EVs that can be captured.

Supplementary Figure 5. EV recovery relative to EV supply at various EV concentrations. We observed that when lower amounts of EVs were supplied (e.g., 10^8 and 10^9 particles), the recovery was comparable to the input. However, at higher EV concentrations (e.g., 10^{10} particles), the recovery reached a saturation point at around 5×10^9 particles.

Finally, we again thank the reviewer for the very constructive suggestions. We hope that the present submitted manuscript is now satisfactory and answers all the reviewer's concerns.

Reviewer #4 :

In this manuscript, Yokoi et al develop the novel use of cellulose membranes for the capture and downstream analysis of extracellular vesicles from small volumes of tissue and bodily fluids. This concept is novel and could have applications for the use of EVs as biomarkers for many disease states, including cancer as discussed. The use of cellulose membranes for the capture, storage and release is easy in methodology and is therefore a reasonable approach for a clinical setting. While this technology has great potential, I have some reservations about whether the cellulose membranes predominantly capture EVs and therefore question the purity of the preparation during downstream analysis. Whether this is an issue for the overall aim of the technology is a separate problem, it is possible that it does not matter.

Response: We are pleased with the reviewer's impression of our manuscript and acknowledge the insightful suggestions offered. We hope that we have now incorporated the additional data required to address the reviewer's concerns and to improve the quality and scientific merit of our work.

Comment #4-1: The main issue here is that the authors attribute the capture of miRNA specifically to EVs, however this finding is overstated as there are few controls and experimental evidence to support these claims. It is known that miRNA can co-isolate with EVs and lipoproteins. The EV populations isolated are characterised based on size, morphology and surface proteins which is fine, however this does not speak to the purity of the isolated material. The removal of contaminants is discussed, and a 10s was step included, however this step is not comprehensively analysed. Therefore, while an interesting technology, the basic science showing that the cellulose membrane isolates EVs specifically is not strong.

Response: The comment was prompted by our failure to provide sufficient explanations in the Methods and other sections. As mentioned in the subsequent comment, we successfully isolated EVs from the sera of 10 ovarian cancer patients, which was confirmed using NTA, cryo-EM, and membrane protein detection. Furthermore, we demonstrated that EV purity is higher with our CNF sheet method than with ultracentrifugation. While the specific separation of lipoproteins and protein aggregates mixed with EVs remains a challenge that needs to be addressed in future research, it is evident that this method can isolate a larger quantity of EVs compared to ultracentrifugation.

Supplementary Figure 7. Comparison of concentration and purity of EVs extracted using the various methods.

(a) Concentration of EVs extracted from 10 μ L of 0.22 μ m-filtered PBS using the CNF sheet, from 250 μ L of

serum using ultracentrifugation, and from 10 μ L of serum using the CNF sheet. (b) EV purity was calculated as the EV concentration (particles/mL) divided by the protein concentration (μ g/mL). EV concentration was measured by NTA and protein concentration was measured by Qubit.

Comment #4-2: I also find the manuscript difficult to read and analyse because there is insufficient detail in the methods, results and figure legends. There is not enough detail for the work to be reproduced.

Response: We carefully addressed the specific comments received and incorporated them into the manuscript. The revisions encompass various sections, including the Methods section, the main text, and figure captions. We provided detailed explanations and made necessary revisions to ensure the clarity and accuracy of the text.

Specific comments

Comment #4-3: Unclear how supp fig 1 supports the sentence beginning on line 5. I assume this is pertaining to water absorbency. It is unclear what supp figure 1 is showing. Please clarify.

Response: We presented our data on the water contact angle measurements and a photograph of a droplet of distilled water taken 10 seconds after dropping it on the cellulose nanofiber (CNF) sheets. The results clearly demonstrate that increasing the pore size of the CNF sheet to around 300 nm significantly improves its water absorption capability. The results for a dense CNF sheet with a pore size of approximately 4 nm are also provided for comparison. Thus, the porous nanostructure design is critical for water absorption, with the CNF sheets with larger pores more effective at absorbing water.

Comment #4-4: Figure 1a – After washing with PBS for only 10s to remove contaminants, it appears that most of the material (EVs?) evident on the surface of the cellulose nanofiber is removed in the SEM image. Also are the SEM images serum EVs? Please update manuscript, methods and figure legend with more information on the SEM. Furthermore, the SEM image of the 5 min wash seem like the fibres are much smaller than the untreated (first image), can the authors comment on this?

Response: The SEM image in Figure 1a was obtained from the CNF sheet with saliva sample. The figure caption has been modified to reflect this change as follows: (a) **Conceptual diagram of the CNF operating principle and the corresponding FE-SEM image obtained using a saliva sample.**

In the text, the following sentence was added to indicate that the results shown in Figure 1a were obtained from the saliva sample: “Finally, EVs were recovered from the fully opened pores of the CNF sheet by immersing it in PBS for 5 minutes. Figure 1a shows a representative FE-SEM image set obtained from a saliva sample.”

For consistency, the following text was added to the Methods section to describe the EV extraction process for each of the body fluids: “**EV extraction from body fluids using CNF sheets.** A volume of 10 μ L of the body fluid sample was introduced on a CNF sheet (10 mm \times 10 mm). The fluid was evenly spread on the sheet using

the a micropipette tip. The CNF sheet was stored and air dried at room temperature for 7 days. Subsequently, the dried CNF sheet was immersed in 1 mL of 0.22 μm -filtered PBS (pH 7.2, Invitrogen) in a Petri dish and washed for 10 seconds. This washing step aimed to remove any excess EVs adhering on the surface of the CNF sheet. Using tweezers, the washed CNF sheet was placed in a 1.5-mL tube (Proteosave SS, Sumitomo Bakelite) filled with 1 mL of 0.22 μm -filtered PBS was used. The tube containing the CNF sheet was then vortexed for 30 seconds. This step was performed to facilitate the extraction of EVs from the CNF sheet into the PBS solution. After 5 minutes, the CNF sheet was removed, leaving the EVs in the PBS solution.”

Furthermore, we clarify that washing does not change the fiber diameter as cellulose nanofibers do not dissolve. The SEM image in Figure R1 (review only) shows some variation in the fiber diameter, while the SEM in Figure 1a incidentally displays thicker fibers.

Figure R1. SEM image of the fabricated CNF sheet.

Comment #4-5: Figure 1d - What do the error bars represent? Is this data from one experiment with several reads or several experiments. Which nanoparticle tracking instrument was used; this is not included in the methodology nor is it in a separate NTA specific section. Supp figure 2a - is this sample the 10s wash step or just EV sheets treated with PBS? I see that there is possibly some EVs removed in the 10s wash step by SEM, has this been quantified by NTA. Where is the evidence that contaminants are removed in the wash step?

Response: Figure 1d shows the size distribution of EVs extracted from the sera of 10 ovarian cancer patients. The error bars represent standard deviation from 10 independent experiments. NanoSight was used for the measurements. The sample in Supp Figure 2a shows the contaminant recovered from 10 μL of 0.22 μm -filtered PBS using CNF sheet. This indicates the debris left from the CNF sheet preparation in 0.22 μm filtered PBS. Kindly note the difference in the scale of the vertical axis, which is 1000-fold different between Fig. 1d and Supp Fig. 2a.

The following text has been added to the Methods section: “**Characterization of EVs extracted from body fluids using CNF sheets.** After the EVs were extracted, their concentration and size were analyzed using a nanoparticle tracking analysis (NTA) instrument (NanoSight LM10 HS; Malvern Panalytical, Ltd.). Video data were collected five times, each with a duration of 60 seconds. The camera level and detection threshold were

set to 15 and 5, respectively. NanoSight NTA 3.2 software was used for data analysis.”

After the washing step (third from the left) in Figure 1a, microRNAs were extracted by adding lysis buffer. During the washing step, EVs adhering to the surface of the CNF sheet were removed. However, it was difficult to quantify the EVs removed during this step because the detection limit of NTA is lower than the detection limit required for these measurements.

The following text has been also added to the Methods section: “**RNA extraction from body fluids using CNF sheets.** The total RNA was extracted by using an RNA purification kit (Norgen Biotek Corp.) following the manufacturer’s instructions. CNF sheets were washed by immersion in 0.22 μm -filtered PBS in a Petri dish for 10 seconds and placed in 1.5 mL of Lysis Buffer A (Norgen Biotek Corp.). In 5-mL tubes (WATSON), 10 μL of β -mercaptoethanol was added to 1 mL of the lysed solution. The mixture was vortexed for 30 seconds and left to stand at room temperature for 5 minutes. After 5 minutes, the CNF sheet was removed. Next, 1.5 mL of 96–100% ethanol was added and the solution was vortexed for 15 seconds. An aliquot of 650 μL of the solution was added to the Norgen Kit’s column and centrifuged at 8,000 rpm for 1 minute to discard the waste solution. This step was repeated until all of the sample was processed (approximately 5 times). The column was washed with 400 μL of wash solution prepared by mixing 18 mL of Wash Solution A (Norgen Biotek Corp.) and 42 mL of 96–100% ethanol. The column was then centrifuged at 14,000 rpm for 1 minute, and the waste solution was discarded. This washing step was repeated three times. Dry spinning was performed at 14,000 rpm for 2 minutes. The column was placed on a new tube, and 50 μL of Elution Solution A (Norgen Biotek Corp.) was added. The column was centrifuged at 2,000 rpm for 2 minutes and 14,000 rpm for 2 minutes to elute the RNA. The RNA concentration was measured using Qubit (ThermoFisher).”

Comment #4-6: It is unclear how EV isolation was performed. There is no mention of volumes for each of the wash steps (one to remove contaminants and one to elute EVs) or filtration steps (see next comment) in the methods section. The figure legends are extremely brief. Figure 1e compares ‘PBS’ and ‘Serum’ in the figure, again this is not further clarified in the figure legend which states that this shows the size distribution of contaminants recovered from “0.22 μm filtered PBS” and “EVs recovered from 10 μL of serum”. The same issue in Fig 1g, unclear, and now one of the samples is called ‘No EVs’. Is this just 0.22 μm filtered PBS treated EV sheets which are then washed 10s then eluted for 5 mins? Please use the same terminology throughout and explain what it means throughout the manuscript (results, figures, methods).

Response: As mentioned earlier, the following items have been added to the Methods section:

“**EV extraction from body fluids using CNF sheets.** A volume of 10 μL of the body fluid sample was introduced on a CNF sheet (10 mm \times 10 mm). The fluid was evenly spread on the sheet using the a micropipette tip. The CNF sheet was stored and air dried at room temperature for 7 days. Subsequently, the dried CNF sheet was immersed in 1 mL of 0.22 μm -filtered PBS (pH 7.2, Invitrogen) in a Petri dish and washed for 10 seconds. This washing step aimed to remove any excess EVs adhering on the surface of the CNF sheet. Using tweezers, the washed CNF sheet was placed in a 1.5-mL tube (Proteosave SS, Sumitomo Bakelite) filled with 1 mL of

0.22 μm -filtered PBS was used. The tube containing the CNF sheet was then vortexed for 30 seconds. This step was performed to facilitate the extraction of EVs from the CNF sheet into the PBS solution. After 5 minutes, the CNF sheet was removed, leaving the EVs in the PBS solution.”

“**Characterization of EVs extracted from body fluids using CNF sheets.** After the EVs were extracted, their concentration and size were analyzed using a nanoparticle tracking analysis (NTA) instrument (NanoSight LM10 HS; Malvern Panalytical, Ltd.). Video data were collected five times, each with a duration of 60 seconds. The camera level and detection threshold were set to 15 and 5, respectively. NanoSight NTA 3.2 software was used for data analysis.”

“**RNA extraction from body fluids using CNF sheets.** The total RNA was extracted by using an RNA purification kit (Norgen Biotek Corp.) following the manufacturer’s instructions. CNF sheets were washed by immersion in 0.22 μm -filtered PBS in a Petri dish for 10 seconds and placed in 1.5 mL of Lysis Buffer A (Norgen Biotek Corp.). In 5-mL tubes (WATSON), 10 μL of β -mercaptoethanol was added to 1 mL of the lysed solution. The mixture was vortexed for 30 seconds and left to stand at room temperature for 5 minutes. After 5 minutes, the CNF sheet was removed. Next, 1.5 mL of 96–100% ethanol was added and the solution was vortexed for 15 seconds. An aliquot of 650 μL of the solution was added to the Norgen Kit’s column and centrifuged at 8,000 rpm for 1 minute to discard the waste solution. This step was repeated until all of the sample was processed (approximately 5 times). The column was washed with 400 μL of wash solution prepared by mixing 18 mL of Wash Solution A (Norgen Biotek Corp.) and 42 mL of 96–100% ethanol. The column was then centrifuged at 14,000 rpm for 1 minute, and the waste solution was discarded. This washing step was repeated three times. Dry spinning was performed at 14,000 rpm for 2 minutes. The column was placed on a new tube, and 50 μL of Elution Solution A (Norgen Biotek Corp.) was added. The column was centrifuged at 2,000 rpm for 2 minutes and 14,000 rpm for 2 minutes to elute the RNA. The RNA concentration was measured using Qubit (ThermoFisher).”

EV isolation was achieved through a series of steps involving body fluid absorption into the CNF sheet, drying, washing, and recovery. Initially, the pore size of the CNF sheet enabled the uptake of substances within the diameter range of 4 to 300 nm through capillary force. After storing the CNF sheet at room temperature for 7 days, the CNF fibers condensed, reducing the pore size to approximately 10 nm. During this stage, EVs and lipoproteins containing phosphate groups possibly interacted with the cellulose nanofibers through hydrogen bonding. The CNF sheets underwent a washing process by immersion in 1 mL of 0.22- μm filtered PBS for 10 seconds. The IR spectrum’s peak shifted (**Supplementary Fig. 3**), indicating that the phosphate groups were more hydrated and interacted via hydrogen bonds with the CNF sheet. Consequently, during the washing process, EVs and lipoproteins were desorbed from the CNF sheet. However, the washing process, which lasted for only 10 seconds, resulted in an expansion of the pore size to around 20 nm, thereby retaining EVs of sizes 30–200 nm inside the CNF sheet. Finally, during the extraction process, EVs were removed by immersing the CNF sheet in 1 mL of 0.22- μm filtered PBS for 5 minutes, effectively increasing and reverting the pore size of the CNF sheet to the range of 4–300 nm.

Supplementary Figure 3. The infrared (IR) spectra were obtained for the following samples: freeze-dried lipid nanoparticles, lipid nanoparticles dispersed in ultrapure water, and lipid nanoparticles on the CNF sheet. The vertical dashed line indicates the wavenumber of 1047 cm^{-1} .

In addition, the caption in Figure 1 has been added and modified as follows:

Figure 1. CNF sheets capture intact EVs. (a) Conceptual illustrations of the operating principle of CNF sheets and corresponding FE-SEM images obtained using a saliva sample. The top row shows an overhead view of the CNF sheet, representing the supply of fluid by pipette, storage and drying for 7 days, washing by immersion in PBS for 10 seconds, and EV extraction by immersion in PBS for 5 minutes. The middle row shows a magnified view of the CNFs. The bottom row shows SEM images during each experimental process when saliva was used. (b) Photograph of EV sheet (~3 inches in diameter) at the time of production (upper row), before use (middle row) cut into 1 cm squares, and after water absorption and drying (lower row). (c) Pore size distribution before supplying body fluids, after drying, after PBS washing for 10 seconds, and after PBS immersion for 5 minutes using the mercury intrusion porosimetry method. (d) Size distribution of EVs recovered from 10 μL of serum using EV sheets. Sera from 10 ovarian cancer patients were used, and error bars indicate standard deviations (SD) for a series of independent measurements ($n = 10$). (e) Size distribution of contaminants recovered from 0.22 μm -filtered PBS and EVs recovered from 10 μL of serum using CNF sheets. Each dot indicates the respective data value, and error bars indicate the SD of 10 measurements. PBS indicates 10 μL of 0.22 μm -filtered PBS; serum indicates 10 μL of serum from ovarian cancer patients pipetted onto CNF sheets, dried, stored, washed, recovered, and measured by NTA. (f) A Cryo-EM image of EVs recovered from serum using CNF sheets. (g) Detection of CD63 with fluorescence-labeled antibody using a well plate and a plate reader. CD63 detection of contaminants recovered from 10 μL of 0.22 μm -filtered PBS (denoted as PBS) and EVs recovered from 10 μL of serum (denoted as serum EVs) using CNF sheets. (h) Exoview detection of EVs

recovered from 10 μ L of serum using CNF sheets. CD63, CD9, and CD81 levels below the line indicate captured antibodies, while CD63, CD9, CD81, and IgG levels above the line indicate detected antibodies. Each dot represents data for each ovarian cancer patient, and error bars represent the SD of a series of independent experiments ($n = 10$).

In addition, we used two different methods for membrane protein detection on EVs and added them to Methods. **Fluorescence detection of CD63 on EV by a plate reader.** A volume of 300 μ L of EVs extracted with 0.22 μ m filtered-PBS were placed in the wells of a 24-well plate. The plate was covered and incubated at room temperature for 20–24 hours. After incubation, the solutions were removed from the wells and 300 μ L of blocking agent (consisting of 90 mL of 0.22 μ m filtered-PBS, 10 mL of Thermo scientific Bloker BSA [10%] in PBS, and 0.5 mL of Tween20 [Funakoshi]) was added. The plates were incubated at room temperature for 1 hour. The blocking agent was removed and the wells were washed twice with 300 μ L of 0.22 μ m filtered-PBS. To each well, 150 μ L of 50 \times diluted anti-CD63 antibody (1 mg/mL, Cosmo Bio) was added and the wells were incubated at room temperature for 1 hour. After incubation, each well was washed with 150 μ L of 0.22- μ m-filtered-PBS. A volume of 150 μ L with 50 \times diluted FITC-labeled anti-IgG (Cosmo Bio Goat anti-mouse IgG antibody) was added to each well. The plates were cover with aluminum to block light and incubated at room temperature for 1 hour. The FITC-labeled anti-IgG solution was removed and the wells were washed thrice with 150 μ L 0.22- μ m-filtered-PBS. Finally, 150 μ L of 0.22- μ m-filtered-PBS was added and the samples were measured using a plate reader (Tecan, Spark, Ex485 nm/Em 535 nm).

Detection of CD63, CD9, and CD81 on EVs by Exoview. Exoview was performed following the manufacturer's instructions as described briefly below. First, the required number of chips from the HumanTetraspanin Kit and Mouse tetraspanin Kit were pre-scanned (NanoView Biosciences). The chips were taken out of the refrigerator and kept at room temperature for 15 minutes before use. The pre-scanned Chip was placed into the center of a 24-well plate using special tweezers. One chip was placed per well, with the Chip ID facing up and ensuring no contact with the wall of the well. The sample was diluted to a concentration of 10^6 – 10^8 particles/mL with incubation solution (10 \times diluted). A volume of 50–70 μ L of the diluted sample was introduced on the Chip, making sure the pipette tip did not touch the Chip. The 24-well plate was sealed and incubated at room temperature for 16 (\pm 1) hours. After incubation, the Chip was washed with 1000 μ L of 10 \times diluted Solution A by running it along the wall surface to avoid direct contact with the Chip. The Chip was shaken with a special Shaker at 500 rpm for 3 minutes, repeating this step three times. The antibodies (CD63, CD9, CD81) were diluted using the blocking solution. For example, for a 300 μ L adjustment, blocking solution: 300 μ L – (0.6 μ L \times 3) + CD63 0.6 μ L + CD9 0.6 μ L + CD81 0.6 μ L. After the 3rd wash, 750 μ L from each well was discarded and 250 μ L of the diluted antibody solution was added. The plate was covered with aluminum foil and shaken for 1 hour. Next, 500 μ L of 10 \times diluted Solution A was added to each well to make the total volume 1000 μ L. A volume of 750 μ L from each well was discard and 750 μ L of 10 \times diluted Solution A was added to each well. The plate was shaken again. A volume of 750 μ L was discarded, 750 μ L of Solution B was added, and the plate was shaken. This was repeated three times. Subsequently 750 μ L of Solution B was removed,

750 μL of DI Water (MilliQ is fine) was added, and the plate was shaken. Two 10–15 cm Petri dishes containing Place MilliQ water was prepared for washing the Chip. Removal of the Chip from the 24-well plates was done horizontally to avoid air contact. The Chip was washed in the Petri dish while turning it horizontally about 10 times. The Chip was removed in the same manner and washed in the other Petri dish. The Chip was slowly removed from the Petri dish at an angle of 45° . Finally, the Chip was scanned and analyzed with the analysis software.

Comment #4-7: It appears that the 10s PBS wash step after serum treatment is never analysed for EVs or contaminants (lipoproteins/proteins/miRNA) and there is little evidence to support the methodology. Furthermore, there is no analysis for contaminants in the 5 min eluate that contains the EVs. Serum is a complex biofluid and isolation of EVs without contaminants difficult. This is a major flaw and does not adhere to the MISEV guidelines which the authors are aware.

Response: The current paper focuses on the *in vivo* analysis of miRNAs on tumor tissue surfaces. Figure 2 presents the results demonstrating the decrease in the concentration of synthetic miRNAs due to the washing process. Similar to the previous analysis of IR spectra, miRNAs would bind to the CNF sheet through hydrogen bonding. Specifically, 10 μL of 50 pM synthetic miRNA (with an NGS read count of 181) was dropped onto the CNF sheet, and after the washing process, the recovered amount of synthetic miRNA was found to be 1.2 pM by qPCR (with an NGS read count of 11). This method appears to be sufficient for miRNA analysis in EVs. To assess EV purity, which is calculated by dividing the EV concentration by the protein concentration, we compared the EV purity extracted from sera of 10 ovarian cancer patients using CNF sheets with that obtained through ultracentrifugation (**Supplementary Fig. 7**). As previously mentioned, achieving complete separation of EVs free from foreign substances is challenging. The CNF sheet method showed higher EV concentration and purity compared to the ultracentrifugation method. While it is possible to analyze foreign substances in general by measuring protein concentrations, it is recommended that future immune-analysis be performed to determine individual concentrations of specific components, such as lipoproteins.

Supplementary Figure 7. Comparison of concentration and purity of EVs extracted using the various methods. (a) Concentration of EVs extracted from 10 μL of 0.22 μm -filtered PBS using the CNF sheet, from 250 μL of serum using ultracentrifugation, and from 10 μL of serum using the CNF sheet. (b) EV purity was calculated as

the EV concentration (particles/mL) divided by the protein concentration ($\mu\text{g/mL}$). EV concentration was measured by NTA and protein concentration was measured by Qubit.

Comment #4-8: Little information is given about the use of ExoView. Is this used in Fig 1h and g, if so why are the units plotted differently in the two graphs? The signal is lower for the CD63 antibody across the board (below isotype controls), yet there is comparable signal for CD9 and CD81 when CD63 is used to capture the EVs. This is contradictory and does not support an enrichment in CD63. However, the authors state that the small EVs isolated can be categorised as CD63-positive small EVs, and the majority could be exosomes. However, the EVs are more CD81 and CD9 positive than CD63. There is no evidence that the EVs isolated are exosomes, they are a heterogeneous population of EVs. For examples of enrichment using ExoView see Breitwieser, K.; Koch, L.F.; Tertel, T.; Proestler, E.; Burgers, L.D.; Lipps, C.; Adjaye, J.; Fürst, R.; Giebel, B.; Saul, M.J. Detailed Characterization of Small Extracellular Vesicles from Different Cell Types Based on Tetraspanin Composition by ExoView R100 Platform. *Int. J. Mol. Sci.* 2022, 23, 8544. <https://doi.org/10.3390/ijms23158544>.

Response: We apologize for any misunderstanding caused by the lack of description in the Methods section. Figure 1g presents the comparison data between serum and 0.22 μm -filtered PBS using immunoassay with fluorescence-labeled antibodies, while Figure 1h displays the expression data of CD63, CD9, and CD81 for EVs obtained from sera of 10 ovarian cancer patients using ExoView. We have also taken your suggestion into account and removed the word “exosome” from the text, as it is indeed a heterogeneous population of EVs.

Comment #4-9: Figure 2 figure legend has b and f incorrectly annotated.

Response: Thank you for your comments. I have made the necessary corrections to the captions of Figure 2b and Figure 2f.

Comment #4-10: The analysis of the removal of free miRNA is not convincing. While it has been shown that EVs contain certain RNAs, it has also been shown that miRNA co-isolate with EVs and lipoproteins. The use of free miRNA as a control does not comprehensively show that the miRNA detected here are in fact inside the EVs isolated on the EV sheets. It would be better to analyse the 10s wash for miRNA after serum treatment. Furthermore, the authors could confirm that the miRNA are ‘inside the EVs’ by comparing the eluate of the 5 min incubation in lysis buffer with that of PBS.

Response: Considering the distribution of EV sizes obtained from the sera of 10 ovarian cancer patients in the 50–300 nm range and the purity of EVs recovered, we are confident that we can detect miRNAs in EVs with fewer mixed lipoproteins compared to serial ultracentrifugation. By using conventional EV isolation methods, it is very challenging to obtain perfectly pure EVs without other contaminants. At the least, our method can isolate EVs with better performance compared to serial ultracentrifugation. Regarding lipoproteins, many functions of miRNA regarding intracellular transportation by HDL⁴ were reported, and LDL can also transport

miRNA in body fluids⁵. However, the size range of these two dominant lipoproteins is around 10 nm, and they are too small to be captured in the CNF sheet. In addition, there are several reported miRNAs, including has-miR-223, has-miR-24, has-miR-135*, has-miR-92a, has-miR-486, has-miR-92a, has-miR-146a, or has-miR-33, which are known to be associated with LDL/HDL. However, these miRNAs were not detected in this study. Furthermore, it is essential to note that there are no direct results indicating complete removal of lipoproteins. Therefore, we have added a discussion that includes the possibility of mixed lipoproteins in the revised manuscript; One of the limitations of EV capturing by CNF sheet is the difficulty of proving how specifically the sheet isolates EVs from biological fluids, and the samples can possibly include non-EV contaminations, such as mixed lipoproteins. This issues also exist in all conventional methods, including serial centrifugation, density gradient centrifugation, size-exclusion chromatography, or any other commercially available isolation kits. It remains technically challenging to prove how much lipoproteins are contaminated in the samples and how to exclude them. Due to the uniqueness of the CNF sheet method, comparing it to conventional methods is also challenging. However, we demonstrated that the sheet can isolate EVs with better performance than serial ultracentrifugation, which is a gold standard method. Regarding lipoproteins, many functions of miRNA regarding intracellular transportation by HDL³² were reported, and LDL can also transport miRNA in body fluids³³. However, the size range of these two dominant lipoproteins is around 10 nm, and they are too small to be captured in the CNF sheet. In addition, there are several reported miRNAs, including has-miR-223, has-miR-24, has-miR-135*, has-miR-92a, has-miR-486, has-miR-92a, has-miR-146a, and has-miR-33, which are known to be associated with LDL/HDL. However, these miRNAs were not detected in this study.

Comment #4-11: All the in vivo work is impressive and interesting, however, again it is unclear whether the results can be specifically attributed to EVs. Certainly, the isolates contain EVs but could also contain lipoproteins, protein aggregates, and associated miRNA.

Response: We acknowledge that the extract could have also contained lipoproteins, protein aggregates, and associated miRNAs. As mentioned above, we have added a discussion in the manuscript that includes the possibility of mixed lipoproteins.

Finally, we again thank the reviewer for the very constructive suggestions. We hope that the present submitted manuscript is now satisfactory and answers all the reviewer's concerns.

References

- 1 Saito, T., Kuramae, R., Wohler, J., Berglund, L. A. & Isogai, A. An ultrastrong nanofibrillar biomaterial: the strength of single cellulose nanofibrils revealed via sonication-induced fragmentation. *Biomacromolecules* **14**, 248-253, doi:10.1021/bm301674e (2013).
- 2 Kobayashi, Y., Saito, T. & Isogai, A. Aerogels with 3D ordered nanofiber skeletons of liquid-crystalline nanocellulose derivatives as tough and transparent insulators. *Angew Chem Int Ed Engl* **53**, 10394-10397, doi:10.1002/anie.201405123 (2014).
- 3 Li, T. *et al.* Developing fibrillated cellulose as a sustainable technological material. *Nature* **590**, 47-56, doi:10.1038/s41586-020-03167-7 (2021).
- 4 Vickers, K. C., Palmisano, B. T., Shoucri, B. M., Shamburek, R. D. & Remaley, A. T. MicroRNAs are transported in plasma and delivered to recipient cells by high-density lipoproteins. *Nat Cell Biol* **13**, 423-433, doi:10.1038/ncb2210 (2011).
- 5 Michell, D. L. & Vickers, K. C. Lipoprotein carriers of microRNAs. *Biochim Biophys Acta* **1861**, 2069-2074, doi:10.1016/j.bbali.2016.01.011 (2016).

REVIEWERS' COMMENTS

Reviewer #1 (Remarks to the Author):

I have no further comment. The concerns have been revised or opportunely justified.

Reviewer #2 (Remarks to the Author):

I appreciate the time and effort to address my concerns regarding this paper. The responses adequately addressed my concerns. I agree the rewording of the discussion seems more appropriate in the current version.

If I could make one more minor suggestion. I recognize that the authors are appropriately using a scientific writing service to assist with the preparation of this manuscript. However, prior to publication, I would encourage this service to check grammar one more time. For instance, on Page 11 Line 36, the word "issues" should be replaced with "issue", singular form. However, if the authors are referring to several issues, "this" should be changed to "these" in line 36.

Best of luck with further development of this technology.

Reviewer #3 (Remarks to the Author):

Dear authors,

corrected manuscript is now prepared in the way that it could be published in the present form.

All the best!

Reviewer #4 (Remarks to the Author):

Thank you to the authors for updating the manuscript with additional data and much more detailed methodology. I believe the data is easier to interpret and reproduce with the amendments. While the authors have not directly shown what proportion of the captured material is EV and non-EV bound, they have explained why they are confident that the majority of the material captured is EVs and why this is the source of miRNA. Therefore, as the data supports their reasoning, I am satisfied with their explanation and updated discussion regarding EV purity and source of miRNA. I believe that this novel methodology for capturing biomarkers of cancer is of significant value and merit and recommend it for publication in Nature Communications.

Response to Reviews

Reviewer #1 (Remarks to the Author):

I have no further comment. The concerns have been revised or opportunely justified.

Response: Thank you very much for reviewing our responses, and we sincerely appreciate your helpful comments to improve this work.

Reviewer #2 (Remarks to the Author):

I appreciate the time and effort to address my concerns regarding this paper. The responses adequately addressed my concerns. I agree the rewording of the discussion seems more appropriate in the current version.

If I could make one more minor suggestion. I recognize that the authors are appropriately using a scientific writing service to assist with the preparation of this manuscript. However, prior to publication, I would encourage this service to check grammar one more time. For instance, on Page 11 Line 36, the word "issues" should be replaced with "issue", singular form. However, if the authors are referring to several issues, "this" should be changed to "these" in line 36.

Best of luck with further development of this technology.

Response: Thank you very much for reviewing our responses, and we carefully rechecked the manuscript, especially in grammar. We sincerely appreciate your helpful comments to improve this work and proudly move on to the next developmental phase.

Reviewer #3 (Remarks to the Author):

Dear authors,

corrected manuscript is now prepared in the way that it could be published in the present form.

All the best!

Response: Thank you very much for reviewing our responses, and we sincerely appreciate your helpful comments to improve this work.

Reviewer #4 (Remarks to the Author):

Thank you to the authors for updating the manuscript with additional data and much more detailed methodology. I believe the data is easier to interpret and reproduce with the amendments. While the authors have not directly shown what proportion of the captured material is EV and non-EV bound, they have explained why they are confident that the majority of the material captured is EVs and why this is the source of miRNA. Therefore, as the data supports their reasoning, I am satisfied with their explanation and updated discussion regarding EV purity and source of miRNA. I believe that this novel methodology for capturing biomarkers of cancer is of significant value and merit and recommend it for publication in Nature Communications.

Response: Thank you very much for reviewing our responses, and we sincerely appreciate your helpful comments to improve this work.